# MODEL-BASED DECENTRALIZED POLICY OPTIMIZATION

## ABSTRACT

Decentralized policy optimization has been commonly used in cooperative multi-agent tasks. However, since all agents are updating their policies simultaneously, from the perspective of individual agents, the environment is non-stationary, resulting in it being hard to guarantee monotonic policy improvement. To help the policy improvement be stable and monotonic, we propose model-based decentralized policy optimization (MDPO), which incorporates a latent variable function to help construct the transition and reward function from an individual perspective. We theoretically analyze that the policy optimization of MDPO is more stable than model-free decentralized policy optimization. Moreover, due to non-stationarity, the latent variable function is varying and hard to be modeled. We further propose a latent variable prediction method to reduce the error of latent variable function, which theoretically contributes to the monotonic policy improvement. Empirically, MDPO can indeed obtain superior performance than model-free decentralized policy optimization in a variety of cooperative multi-agent tasks.

## 1 INTRODUCTION

Decentralized multi-agent reinforcement learning (MARL) has been commonly used in practice for cooperative multi-agent tasks, *e.g.*, traffic signal control (Wei et al., 2018), unmanned aerial vehicles (Qie et al., 2019), and IoT (Cao et al., 2020), where global information is inaccessible. Independently performing policy optimization using local information, *e.g.*, independent PPO (Schulman et al., 2017) (IPPO), is one of the most straightforward methods for decentralized MARL. Recent empirical studies (de Witt et al., 2020; Yu et al., 2021a; Papoudakis et al., 2021) demonstrate that IPPO performs surprisingly well in several cooperative multi-agent benchmarks, which shows great promise for fully decentralized policy optimization.

However, since all agents are updating policies, from the perspective of an individual agent, the environment is non-stationary (Zhang et al., 2019). Thus, the monotonic policy improvement, which can be achieved by policy optimization in single-agent settings (Schulman et al., 2015; 2017), may not be guaranteed in decentralized MARL. Concretely, in policy optimization, the state visitation frequency is assumed to be stationary since the agent policy is limited to slight updates, which is necessary to guarantee monotonic policy improvement (Schulman et al., 2015). However, in decentralized multi-agent settings, as all agents are updating policies simultaneously, the state visitation frequency will change greatly, which contradicts the fundamental assumption of policy optimization, thus the monotonic improvement of policy optimization may not be preserved.

To address this problem, we resort to exploiting the environment model to stabilize the state visitation frequency and help monotonic policy improvement. However, learning an environment model in decentralized settings is non-trivial, since the information of other agents, *e.g.*, other agents' policies, is not observable and changing. Therefore, we introduce a latent variable to help distinguish different transitions resulting from the unobservable information. And then we build an environment model for each agent, which contains a transition function, a reward function, and a latent variable function that learns the latent variable given observation. The agents are trained using independent policy optimization methods, *e.g.*, TRPO (Schulman et al., 2015) or PPO (de Witt et al., 2020), on both the experiences generated by the environment model and collected in the environment.

Since the environment is non-stationary, the latent variable function is also varying during learning. We theoretically show that independently performing policy optimization on experiences generated

by the environment model with the varying latent variable function can obtain more stationary observation visitation frequency than on the experiences collected in the non-stationary environment. Thus, independent policy optimization goes more stable on the environment model.

Moreover, to obtain monotonic improvement, the gap between the return of interacting with the environment and the return predicted by the environment model should be small. We theoretically analyze that the return gap is bounded by the prediction error of the latent variable function. As the latent variable function is varying due to non-stationarity, to minimize the prediction error, we propose a latent variable prediction method that uses the historical variables to predict the future variable. Thus, the latent variable prediction can reduce the return gap and help the monotonic policy improvement.

The proposed algorithm, ***model-based decentralized policy optimization*** (**MDPO**), is theoretically grounded and empirically effective for fully decentralized learning. We evaluate MDPO on a variety of cooperative multi-agent tasks, *i.e.*, a stochastic game, multi-agent particle environment (MPE) (Lowe et al., 2017), and multi-agent MuJoCo (Peng et al., 2021a). MDPO outperforms the model-free independent policy optimization baseline, and the proposed latent variable prediction additionally obtains performance gain, verifying that MDPO can help stable and monotonic policy improvement in fully decentralized learning.

## 2 PRELIMINARIES

**Dec-POMDP.** A cooperative multi-agent task is generally modeled as a decentralized partially observable Markov decision process (Dec-POMDP) (Oliehoek & Amato, 2016). Specifically, a Dec-POMDP is defined as a tuple $G = \{\mathcal{S}, \mathcal{I}, \mathcal{A}, \mathcal{O}, \Omega, P, R, \gamma\}$. $\mathcal{S}$ is the state space, $\mathcal{I}$ is the set of agents, and $\mathcal{A} = A_1 \times \cdots \times A_{|\mathcal{I}|}$ is the joint action space, where $A_i$ is the action space for each agent $i$. At each state $s$, each agent $i \in \mathcal{I}$ merely gets access to the observation $o_i \in \mathcal{O}$, which is drawn from observation function $\Omega(s, i)$, and selects an action $a_i \in A_i$, and all the actions form a joint action $\boldsymbol{a} \in \mathcal{A}$. The state transitions to next $s'$ according to the transition function $P(s'|s, \boldsymbol{a}) : \mathcal{S} \times \mathcal{A} \times \mathcal{S} \rightarrow [0, 1]$, and all agents receive a shared reward $r = R(s, \boldsymbol{a}) : \mathcal{S} \times \mathcal{A} \rightarrow \mathbb{R}$. The objective is to maximize the expected return $\eta(\boldsymbol{\pi}) = \mathbb{E}[\sum_{t=0}^{\infty} \gamma^t r_t | \rho_0, \boldsymbol{\pi}]$ under the joint policy of all agents $\boldsymbol{\pi}$ and distribution of initial state $\rho_0$, where $\gamma \in [0, 1)$ is the discounted factor. The joint policy $\boldsymbol{\pi}$ can be represented as the product of each agent's policy $\pi_i$. Also we denote $\boldsymbol{\pi}_{-i}$ as the joint policy of all agents except $i$.

**Fully decentralized learning.** We consider the fully decentralized way to solve the Dec-POMDP (Tan, 1993; de Witt et al., 2020), where each agent independently learns a policy and executes actions without communication or parameter sharing in both training and execution phases. Since all agents are updating policies, from the perspective of individual agents, the environment is non-stationary, which fundamentally challenges decentralized learning (Zhang et al., 2019). The existing decentralized MARL methods are limited. Independent Q-learning (IQL) (Tan, 1993) and independent policy optimization, *e.g.*, IPPO (de Witt et al., 2020), are the most straightforward fully decentralized algorithms. Despite good empirical performance (Papoudakis et al., 2021), due to non-stationarity, these methods lack theoretical guarantees. IQL has no convergence guarantee, to the best of our knowledge. Although there has been some study (Sun et al., 2022), IPPO may not guarantee policy improvement by independent policy optimization, since the assumption of stationary state visitation frequency for policy optimization may not hold in fully decentralized settings, which we will discuss in the following.

**Monotonic policy improvement.** In Dec-POMDP, from a centralized perspective, we can obtain a TRPO objective (Schulman et al., 2015) of the joint policy $\boldsymbol{\pi}$ for the monotonic improvement,

$$\eta(\boldsymbol{\pi}^{\text{new}}) - \eta(\boldsymbol{\pi}^{\text{old}}) \geq \sum_s \rho^{\boldsymbol{\pi}^{\text{new}}}(s) \sum_{\boldsymbol{a}} \boldsymbol{\pi}^{\text{new}}(\boldsymbol{a}|s) A^{\boldsymbol{\pi}^{\text{old}}}(s, \boldsymbol{a}) - C \cdot D_{\text{KL}}^{\max}(\boldsymbol{\pi}^{\text{old}} \| \boldsymbol{\pi}^{\text{new}}) \quad (1)$$

$$\gtrapprox \sum_s \rho^{\boldsymbol{\pi}^{\text{old}}}(s) \sum_{\boldsymbol{a}} \boldsymbol{\pi}^{\text{new}}(\boldsymbol{a}|s) A^{\boldsymbol{\pi}^{\text{old}}}(s, \boldsymbol{a}) - C \cdot D_{\text{KL}}^{\max}(\boldsymbol{\pi}^{\text{old}} \| \boldsymbol{\pi}^{\text{new}}), \quad (2)$$

where $\rho^{\boldsymbol{\pi}^{\text{old}}}(s) = \sum_{t=0} \gamma^t \Pr(s_t = s | \boldsymbol{\pi}^{\text{old}})$ is the discounted state visitation frequency given $\boldsymbol{\pi}^{\text{old}}$, similarly for $\rho^{\boldsymbol{\pi}^{\text{new}}}(s)$, $A^{\boldsymbol{\pi}^{\text{old}}}$ is the advantage function under $\boldsymbol{\pi}^{\text{old}}$, $D_{\text{KL}}^{\max}(\boldsymbol{\pi}^{\text{old}} \| \boldsymbol{\pi}^{\text{new}}) = \max_s D_{\text{KL}}(\boldsymbol{\pi}^{\text{old}}(\cdot|s) \| \boldsymbol{\pi}^{\text{new}}(\cdot|s))$, and $C$ is a constant. From (1) to (2) is an approximation or

assumption (Schulman et al., 2015). As $\rho^{\pi^{\mathrm{new}}}$ is *unknown* and the policy is limited to slight updates, $\rho^{\pi^{\mathrm{new}}}$ is approximated by $\rho^{\pi^{\mathrm{old}}}$. However, in fully decentralized MARL, this assumption may not hold, as all agents are updating their policies simultaneously and their joint policy may change significantly especially when the number of agents is large. This will severely affect the performance of independent policy optimization. Although we can constrain the policy update of each agent to be slight like TRPO, this leads to much slower convergence, especially in fully decentralized MARL, where the joint policy has a much larger search space and is merely optimized by independent learning of individual agents.

## 3 MODEL-BASED DECENTRALIZED POLICY OPTIMIZATION

In this paper, we provide a novel perspective and resort to the environment model to bridge the gap between $\rho^{\pi^{\mathrm{new}}}$ and $\rho^{\pi^{\mathrm{old}}}$ for each agent such that the monotonic joint policy improvement can be potentially achieved by fully decentralized policy optimization.

As illustrated in the following, we turn the learning process into a Dyna-style (Sutton, 1990) decentralized model-based method with the green path [1]. Each agent $i$ additionally learns a decentralized model using local information from policy rollout and can optionally perform policy optimization on the experiences from model rollout. When optimizing policy with model rollout, we essentially have ①, which means the state visitation frequency in model rollout ($\rho_{\mathrm{model}}$) is more stable. Thus, the approximation from (1) to (2) becomes acceptable under looser constraints of policy update. Further, we can bound the gap between the returns of policy rollout ($\eta$) and model rollout ($\eta^{\mathrm{model}}$), formally in ②. Once the bound ($\mathcal{B}$) is controllable throughout the learning process, it can potentially guarantee the monotonic improvement of the joint policy in the real environment.

Thus, ① and ② together highlight the potential benefits of incorporating an environment model for decentralized policy optimization. In the following, we discuss how to learn such a decentralized model, theoretically investigate its benefits for decentralized policy optimization, and analyze the return bound for monotonic policy improvement.

### 3.1 LATENT VARIABLE MODEL

In decentralized learning, from the perspective of each agent $i$, the transition function and reward function are respectively,

$$P_i(s'|s,a_i) = \mathbb{E}_{\boldsymbol{a}_{-i} \sim \pi_{-i}} P(s'|s,a,\boldsymbol{a}_{-i}) \quad \text{and} \quad R_i(s,a_i) = \mathbb{E}_{\boldsymbol{a}_{-i} \sim \pi_{-i}} R(s,a,\boldsymbol{a}_{-i}),$$

where $\boldsymbol{a}_{-i}$ denotes the joint action of all agents except $i$. As other agents are also updating their policies, $P_i$ and $R_i$ are varying throughout the learning process, which is the well-known non-stationarity problem. Moreover, as each agent $i$ usually obtains observation instead of state in decentralized learning, the model can only be learned on $(o_i, a_i, o'_i, r)$. Thus, it is challenging to construct an environment model from the perspective of an individual agent.

To build a decentralized environment model, we introduce a latent variable $z_i$, which helps distinguish different transitions resulting from varying unobservable information of the full state and other agents' policies. Then the transition function and the reward function can be redefined as:

$$P_i(o'_i|o_i,a_i,z_i) \quad \text{and} \quad R_i(o_i,a_i,z_i).$$

As we discuss fully decentralized learning, we drop the subscript of $i$ for simplicity in the following.

To model the transition function and the reward function with the latent variable, we define the latent variable function from the perspective of an individual agent, $\psi(z|o)$, which indicates the

---

[1]Related work on model-based MARL can be found in Appendix E. However, none of the existing work considers exploiting the environment model to help fully decentralized policy optimization.

probability of latent variable $z$ given observation $o$. As $z$ is related to the policies of other agents, $\psi(z|o)$ also varies during policy updates. A latent variable model consists of three modules: transition function $P_\theta$, reward function $R_\phi$, and latent variable function $\psi_\omega$, to predict the next observation and reward. As the impact of unobservable information is designed to merely reflect on the latent variable, although other agents update their policies, the transition function and reward function stay constant and only the latent variable function varies. We learn such a model by maximizing the likelihood of experiences of policy rollout $\mathcal{D}$,

$$\max_{\theta,\omega,\phi} \mathbb{E}_{(o,a,o',r)\sim\mathcal{D},z\sim\psi_\omega(\cdot|o)} \left[ P_\theta\left(o'|o,a,z\right) - \left(R_\phi\left(o,a,o',z\right) - r\right)^2 \right]. \tag{3}$$

We examine the correlation between latent variable learned end-to-end and inaccessible information in a simple setting, and the learned latent variable is indeed correlated with the inaccessible information. More details can be found in Appedix B.

Moreover, when using the learned latent variable model to train an agent, we adopt $k$-step branched model rollout in MBPO (Janner et al., 2019) to avoid compounding model error due to long-horizon rollout. Concretely, at each policy update of an agent, we sample $h$-step length experiences $\{(o_1,a_1,o'_1,r_1),\cdots,(o_h,a_h,o'_h,r_h)\}$ from policy rollout $\mathcal{D}$ and perform $k$-step model rollout starting from the last observation $o'_h$ under current policy $\pi$. The policy $\pi$ is updated on the merged $(h+k)$-step experiences $\{(o_1,a_1,o'_1,r_1),\cdots,(o_{h+k},a_{h+k},o'_{h+k},r_{h+k})\}$ by policy optimization, *e.g.*, PPO (Schulman et al., 2015).

## 3.2 STABLE POLICY OPTIMIZATION ON MODEL

Now, we turn to analyze the benefits of such a model-based method over model-free independent policy optimization. We first theoretically analyze that independently performing policy optimization *e.g.,*, TRPO (Schulman et al., 2015) or PPO (Schulman et al., 2017), on the latent variable model can make the learning process more stable.

In decentralized learning, from the perspective of an agent, given the *true* latent variable function $\psi$, the discounted observation visitation frequency of $\mathcal{D}$ obtained by policy rollout is defined as

$$\rho^{\pi,\psi}\left(o\right) = \rho_0^{\pi,\psi}\left(o\right) + \gamma\rho_1^{\pi,\psi}\left(o\right) + \gamma^2\rho_2^{\pi,\psi}\left(o\right) + \cdots,$$

where $\rho_t^{\pi,\psi}\left(o\right) \triangleq Pr(o_t = o)$ and $o_t$ is the observation at timestep $t$ of experience from $\mathcal{D}$. Note that $\rho^{\pi,\psi}\left(o\right)$ is an unbiased estimate of discounted observation visitation frequency when interacting in the environment. Similarly, $\rho^{\pi,\psi_\omega}$ denotes the discounted observation visitation frequency for experiences obtained by model rollout. During the learning process, $\pi^n$ and $\psi^n$ respectively denote the policy and latent variable function after the $n$th policy update. Then, we have the following theorem. All proofs are available in Appendix A.

**Theorem 1.** *Define* $\Delta\rho^n(o) \triangleq \rho^{\pi^n,\psi^n}\left(o\right) - \rho^{\pi^{n-1},\psi^{n-1}}\left(o\right)$, *and denote* $\|\Delta\rho^n\| \triangleq \max_o|\rho^n(o) - \psi^{n-1}(o)|$, *similarly for* $\|\Delta\pi^n\|$ *and* $\|\Delta\psi^n\|$. *It holds that,*

$$\|\Delta\rho^n\| \leq C\left(\mathcal{E}_\pi + \mathcal{E}_\psi\right),$$

*where* $\mathcal{E}_\pi \triangleq \max_n \|\Delta\pi^n\|$, $\mathcal{E}_\psi \triangleq \max_n \|\Delta\psi^n\|$ *and* $C$ *is a constant. Assume* $\psi_\omega^n = (1-\alpha)\psi_\omega^{n-1}+\alpha\psi^n$ *and* $\psi_\omega^0 = \psi^0$ [2]. *It holds that* $\mathcal{E}_\psi > \mathcal{E}_{\psi_\omega}$ *and the bound above is lower when substituting* $\psi$ *with* $\psi_\omega$.

According to Theorem 1, the divergence of discounted observation visitation frequency is bounded by the divergence of policy and latent variable function. Again, the policy divergence can be constrained via policy optimization, like TRPO. Thus, the main difference lies in the divergence of latent variable function. As indicated by Theorem 1, the learned latent variable function $\psi_\omega$ has a smaller divergence between consecutive policy rollouts than the true latent variable function $\psi$. *Therefore, independent policy optimization on experiences generated by the latent variable model can obtain more stationary observation visitation frequency than on the experiences collected in the varying environment, so the learning process of independent policy optimization becomes stable on the model.*

---

[2]Since $\psi$ is varying and $\psi_\omega$ is continuously updated using the experiences from several recent policy rollouts, we use the form of soft-update for the relation between $\psi$ and $\psi_\omega$.

### 3.3 RETURN BOUNDS

We then analyze the bound of return gap between interacting in the environment and interacting with the model. If the return improvement of interacting with the model is higher than the bound, the agent can obtain the monotonic policy improvement when interacting in the environment.

However, the return of interacting in the environment is hard to analyze in decentralized learning since the policies of other agents are inaccessible, we turn to analyze the return in policy rollout, which is an unbiased estimate of expected return in the environment.

Several bounds have been introduced in MBPO (Janner et al., 2019) for the return bound analysis, which however are not sufficient in decentralized learning. Thus, we need to introduce two new bounds that indicate the divergence of latent variable function between consecutive policy rollouts and the error of learned latent variable function. Formally, with $\psi^n$, $\psi_\omega^n$, and $\pi^n$ respectively referring to the true latent variable function, the learned latent variable function, and the policy of the $n$th policy rollout, we denote the bounds as follows:

- reward bound $r_{\max} \triangleq \max_{o,a,z} \max\{R(o,a,z), R_\phi(o,a,z)\}$;
- policy divergence $\epsilon_\pi \triangleq \max_o D_{TV}(\pi\|\pi^n)$, where $D_{TV}$ is total variation distance;
- transition function error $\epsilon_\theta \triangleq \max_t \mathbb{E}_{a_t,z_t \sim (\pi,\psi_\omega^n)} D_{TV}(P(o_{t+1}|o_t,a_t)\|P_\theta(o_{t+1}|o_t,a_t,z_t))$;
- *latent variable function divergence* $\epsilon_\psi \triangleq \max_o D_{TV}(\psi^n\|\psi^{n+1})$;
- *learned latent variable function error* $\epsilon_\omega \triangleq \max_o D_{TV}(\psi^n\|\psi_\omega^n)$.

Additionally, we use several notations to represent different returns. The return in $n$th policy rollout with the true latent variable function $\psi$ is denoted as $\eta(\pi, \psi)$, the return in model rollout with the $n$th learned latent variable function $\psi_\omega$ is denoted as $\eta^{model}(\pi, \psi_\omega^n)$, and the return in $k$-step branched model rollout with $h$-step experiences of $n$th policy rollout is denoted as $\eta^{branch}((\pi^n, \pi), (\psi, \psi_\omega^n))$. Now we analyze the return bound of model rollout and branched model rollout with the newly introduced $\epsilon_\psi$ and $\epsilon_\omega$ in the following two theorems.

**Theorem 2.** *The return gap between $n+1$th policy rollout and model rollout with $n$th learned model is bounded as:*

$$\left|\eta\left(\pi, \psi^{n+1}\right) - \eta^{\mathrm{model}}\left(\pi, \psi_\omega^n\right)\right| \leq \underbrace{\frac{2r_{\max}}{(1-\gamma)^2}\left(\gamma\epsilon_\theta + 2\epsilon_\pi + 2\epsilon_\omega + \epsilon_\psi\right)}_{C(\epsilon_\theta, \epsilon_\pi, \epsilon_\omega, \epsilon_\psi)}.$$

**Theorem 3.** *The return gap between $n+1$th policy rollout and branched model rollout with $n$th learned model is bounded as:*

$$\left|\eta\left(\pi, \psi^{n+1}\right) - \eta^{\mathrm{branch}}\left((\pi^n, \pi), (\psi^n, \psi_\omega^n)\right)\right| \leq C\left(\epsilon_\theta, \epsilon_\pi, \epsilon_\omega, \epsilon_\psi\right).$$

According to Theorem 2 and 3, we can guarantee the monotonic improvement in the environment via improving the return in model rollout or branched model rollout beyond a bound linear to $(\epsilon_\pi, \epsilon_\theta, \epsilon_\omega, \epsilon_\psi)$. In these bounds, $\epsilon_\theta$ and $\epsilon_\omega$ are limited via supervised learning and $\epsilon_\pi$ is constrained by policy optimization. However, $\epsilon_\psi$ is left *unrestricted*. In the following, we try to find a better bound in which all elements are controllable.

### 3.4 LATENT VARIABLE PREDICTION

In order to restrict the impact of divergence of the latent variable function, we introduce one new error bound, which measures the divergence between the learned latent variable function and the true latent variable function in incoming policy rollout. Formally, such an error bound in $n$th policy rollout is defined as:

$$\hat{\epsilon}_\omega \triangleq \max_o D_{TV}\left(\psi^{n+1}\|\psi_\omega^n\right).$$

Now we use $\hat{\epsilon}_\omega$ in place of $\epsilon_\psi$ to analyze the return bound of model rollout and branched model rollout again in the following two theorems.

**Theorem 4.** *The return gap of $n+1$th policy rollout and model rollout with $n$th learned model is bounded as:*

$$\left|\eta\left(\pi, \psi^{n+1}\right) - \eta^{\mathrm{model}}\left(\pi, \psi_\omega^n\right)\right| \leq C\left(\epsilon_\theta, \epsilon_\pi, \hat{\epsilon}_\omega\right).$$

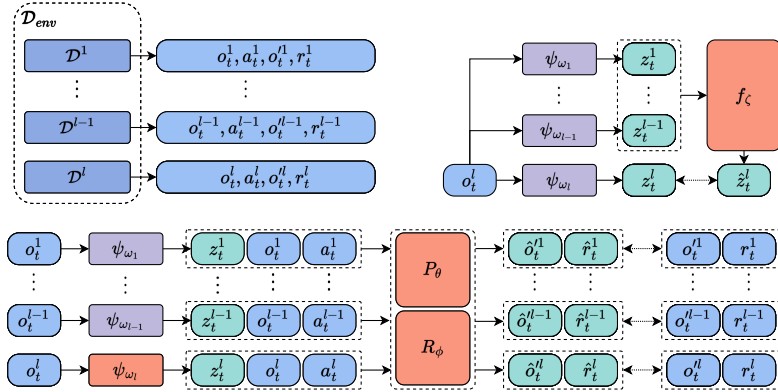

Figure 1: The environment model includes four modules: transition function $P_\theta$, reward function $R_\phi$, latent variable prediction function $f_\zeta$, and latent variable functions $\{\psi_{\omega_1}, \cdots, \psi_{\omega_l}\}$ over $l$ consecutive policy rollouts. For learning, each agent maintains the experiences of $l$ consecutive policy rollouts, $P_\theta$ and $R_\phi$ learn on $l$ consecutive policy rollouts, $\psi_{\omega_l}$ learns on the experiences of $l$th policy rollout, and $f_\zeta$ learns to predict $l$th latent variable given $l-1$ latent variable functions.

**Theorem 5.** *The return gap of $n+1$th policy rollout and branched model rollout with $n$th learned model is bounded as:*

$$\left| \eta\left(\pi, \psi^{n+1}\right) - \eta^{\text{branch}}\left((\pi^n, \pi), (\psi^n, \psi^n_\omega)\right) \right| \leq C\left(\epsilon_\theta, \epsilon_\pi, \epsilon_\omega, \hat{\epsilon}_\omega\right).$$

Now all elements of the bounds are controllable, once we can constrain $\hat{\epsilon}_\omega$ in the learning process. To achieve this, we introduce a latent variable prediction function, which predicts the latent variable distribution given observation $o$ in incoming policy rollout via latent variable distributions of $o$ in the latest $l-1$ policy rollouts. However, as the true latent variable function cannot be obtained directly for an agent, the latent variable prediction function $f_\zeta$ can instead minimize:

$$\max_o D_{TV}\left(\psi_{\omega_l}(o) \| f\left(\psi_{\omega_1}(o), \cdots, \psi_{\omega_{l-1}}(o)\right)\right).$$

With such a latent variable prediction function, $\hat{\epsilon}_\omega$ is controllable.

### 3.5 ALGORITHM

With all the theoretical analysis and discussions above, we are ready to present the learning of *model-based decentralized policy optimization* (MDPO).

As illustrated in Figure 1, the environment model consists of transition function $P_\theta$, reward function $R_\phi$, latent variable prediction function $f_\zeta$, and latent variable functions $\{\psi_{\omega_1}, \cdots, \psi_{\omega_l}\}$ over recent $l$ consecutive policy rollouts. The experiences of the $l$ consecutive policy rollouts $\mathcal{D}_{env} = \{\mathcal{D}^1, \cdots, \mathcal{D}^l\}$ are also stored.

After the latest policy rollout $l$, we update the transition function and reward function, and learn the latent variable function $\psi_{\omega_l}$ of policy rollout $l$ by optimizing the objective:

$$\max_{\theta, \omega_l, \phi} \sum_{j=1}^{l} \left( \mathbb{E}_{(o,a,o',r) \sim \mathcal{D}^j, z \sim \psi_{\omega_j}(o)} \left[ P_\theta\left(o'|o,a,z\right) - \left(R_\phi\left(o,a,z\right) - r\right)^2 \right] \right), \qquad (4)$$

where $\psi_{\omega_l}$ is obtained by updating $\psi_{\omega_{l-1}}$ using $\mathcal{D}^l$, while $P_\theta$ and $R_\phi$ are updated using $\mathcal{D}_{env}$ to make sure they are stable across policy rollouts. Then, the latent variable prediction function $f_\zeta$ is updated using $\mathcal{D}^l$ by optimizing the objective:

$$\max_\zeta \mathbb{E}_{o \sim \mathcal{D}^l, z^1 \sim \psi_{\omega_1}(o), \cdots, z^l \sim \psi_{\omega_l}(o)} \left[ f_\zeta\left(z^l | z^1, \cdots, z^{l-1}\right) \right]. \qquad (5)$$

For model rollout, the model predicts the transition in incoming policy rollout given observation $o$ and action $a$ via $l-1$ latest learned latent variable functions $(\psi_2, \cdots, \psi_l)$ as:

$$z \sim f_\zeta\left(\cdot | z^2 \sim \psi_{\omega_2}(o), \cdots, z^l \sim \psi_{\omega_l}(o)\right), \hat{o}' \sim P_\theta\left(o,a,z\right), \hat{r} = R_\phi\left(o,a,z\right). \qquad (6)$$

Finally, the policy is updated using the branched model rollout by policy optimization, such as PPO or TRPO. We summarize the full learning procedure of MDPO in Algorithm 1.

---

**Algorithm 1. MDPO**

---

 1: **Initiate** $\mathcal{D}_{env} = \{\mathcal{D}^1, \cdots, \mathcal{D}^l\}, \pi, P_\theta, R_\phi, \Psi = \{\psi_{\omega_1}, \cdots, \psi_{\omega_l}\}, f_\zeta$.
 2: **repeat**
 3:     policy rollout in environment and obtain $\mathcal{D}^l$
 4:     optimize $P_\theta, R_\phi$ and $\psi_{\omega_l}$ on $\mathcal{D}_{env}$ with (4)
 5:     optimize prediction function $f_\zeta$ on $\mathcal{D}^l$ with (5)
 6:     obtain branched model rollout $\mathcal{D}_{rollout}$ based on $\mathcal{D}^l$ using $P_\theta, R_\phi, \pi, \Psi$, and $f_\zeta$ with (6)
 7:     optimize policy $\pi$ using $\mathcal{D}_{rollout}$ by PPO or TRPO
 8:     **for** $j \leftarrow 1, \ldots, l-1$ **do**
 9:         $\mathcal{D}^j \leftarrow \mathcal{D}^{j+1}, \psi_{\omega_j} \leftarrow \psi_{\omega_{j+1}}$
10:     **end for**
11: **until** terminate

---

## 4 EXPERIMENTS

For evaluation, we compare MDPO, MDPO without latent variable prediction (denoted by MDPO w/o prediction), and independent PPO (IPPO) (Schulman et al., 2017) on a set of cooperative multi-agent tasks including a stochastic game, multi-agent particle environment (MPE) (Lowe et al., 2017), and multi-agent MuJoCo (Peng et al., 2021b). We do not consider StarCraft multi-agent challenge (SMAC) (Samvelyan et al., 2019), because IPPO has been shown to perform very well in SMAC (de Witt et al., 2020; Papoudakis et al., 2021), close enough to centralized training with decentralized execution methods like QMIX (Rashid et al., 2018) and MAPPO (Yu et al., 2021a). Thus, the gain of MDPO may not be clearly evidenced there.

By experiments, we try to answer the following three questions:

1. *Does the latent variable model help to generate experiences with more stationary observation visitation frequency experimentally?*

2. *Does latent variable prediction help to control $\hat{\epsilon}_\omega$?*

3. *Does MDPO help to improve performance in decentralized learning?*

For a fair comparison, the network architecture and hyperparameters are the same for IPPO and MDPO. The number of environment steps taken in each round (policy rollout, network update) is consistent and thus we compare the performance of methods under the same number of environment steps and policy updates. Note that since we consider fully decentralized learning, for all methods, *agents do not use parameter-sharing.* Indeed, parameter-sharing should not be allowed in decentralized learning (Terry et al., 2020). More details on experiment settings, implementation, and hyperparameters are available in Appendix C.

### 4.1 STOCHASTIC GAME

The stochastic game is a cooperative game with 30 observations (states), 3 agents, and 5 actions for each agent, and every episode consists of 40 steps. The transition function and the shared reward function are randomly generated. The game is chosen to verify our theoretical results.

Figure 2 (left) shows the learning curves of MDPO, MDPO w/o prediction, and IPPO, among which MDPO performs better throughout the learning process. With a finite observation space in this game, we calculate the divergence of observation visitation frequencies ($\|\Delta\rho\|$ in Section 3.2) in consecutive rollouts. Concretely, we calculate the L1 distance of observation visitation frequency over all observations in consecutive rollouts (policy rollouts for IPPO and branched model rollouts for MDPO), and their curves are shown in Figure 2 (mid). We can see that the latent variable model generates experiences with more stationary observation visitation frequency than IPPO, which is consistent with Theorem 1. This may account for the superior performance of MDPO w/o prediction over IPPO.

We also examine how well the latent variable prediction helps to control the prediction error ($\hat{\epsilon}_\omega$). As the real latent variable function is inaccessible, we examine $\hat{\epsilon}_\omega$ by comparing how well the learned environment model predicts with and without latent variable prediction. Specifically, we measure the

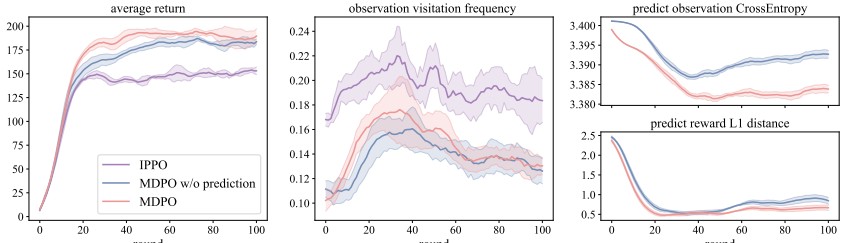

Figure 2: Learning curves of MDPO compared with MDPO w/o prediction and IPPO on the stochastic game: average return (left), observation visitation frequency divergence (mid), and model prediction errors (right). Each round is 1600 environment steps.

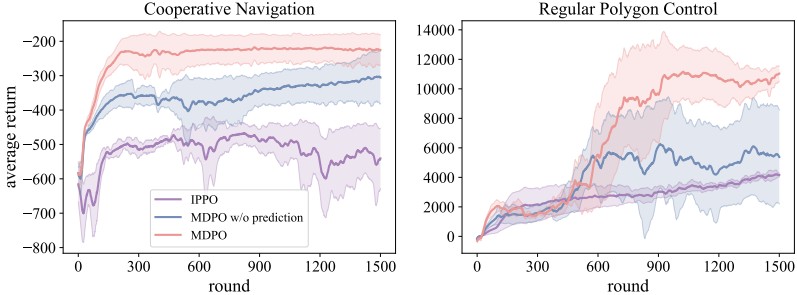

Figure 3: Learning curves of MDPO, MDPO w/o prediction, and IPPO in Cooperative Navigation (left) and Regular Polygon Control (right). Each round is 1280 environment steps.

mean cross-entropy of the next observation distribution predicted by the $n$th learned model and the ground truth in the $n + 1$th round, and the mean L1 distance of predicted reward and ground truth reward. The curves are shown in Figure 2 (right). The lower prediction error of MDPO indicates that latent variable prediction error ($\hat{\epsilon}_\omega$) is controlled at a lower level than without latent variable prediction. Moreover, as shown in Figure 2 (mid), the divergence of observation visitation frequency of MDPO and MDPO w/o prediction are similar but much lower than IPPO, which indicates $\epsilon_\omega$ is still under control in MDPO. This indicates that MDPO can well control both $\epsilon_\omega$ and $\hat{\epsilon}_\omega$.

As MDPO helps to handle non-stationarity in multi-agent settings from the perspective of an individual agent, it will be natural to also apply MDPO to non-stationary single-agent settings. So, we modify this stochastic game into a non-stationary single-agent game and show that MDPO also outperforms the baselines. More details are available in Appendix D.

### 4.2 MPE

MPE is a multi-agent environment with continuous observation. In our MPE tasks, agents observe their own positions, velocity, and others' relative positions. And agents are expected to fulfill a certain goal via controlling their accelerations in every direction which is continuous in our experiments. Two tasks of MPE, 4-agent Cooperative Navigation and 5-agent Regular Polygon Control, are chosen for performance comparison. In 4-agent Cooperative Navigation, 4 agents learn to cooperate to reach 4 landmarks respectively. In 5-agent Regular Polygon Control, 4 agents learn to cooperate with another agent, which is controlled by a fixed policy, aiming to form a regular pentagon, and the reward is given according to the similarity to a regular pentagon.

Figure 3 shows the learning curves of all methods. Generally, MDPO w/o prediction performs better than IPPO, which verifies that the latent variable model can help decentralized policy improvement by making the observation visitation frequency more stationary. And MDPO outperforms MDPO w/o prediction, which verifies latent variable prediction can reduce the gap between the return of interaction and the return predicted by the environment model.

It is worth noting that the unobservable information required to fulfill the goal is at completely different levels in the two tasks. Concretely, acknowledging the general direction of others is enough to decide which landmark to approach in Cooperative Navigation. Yet the precise positions of others

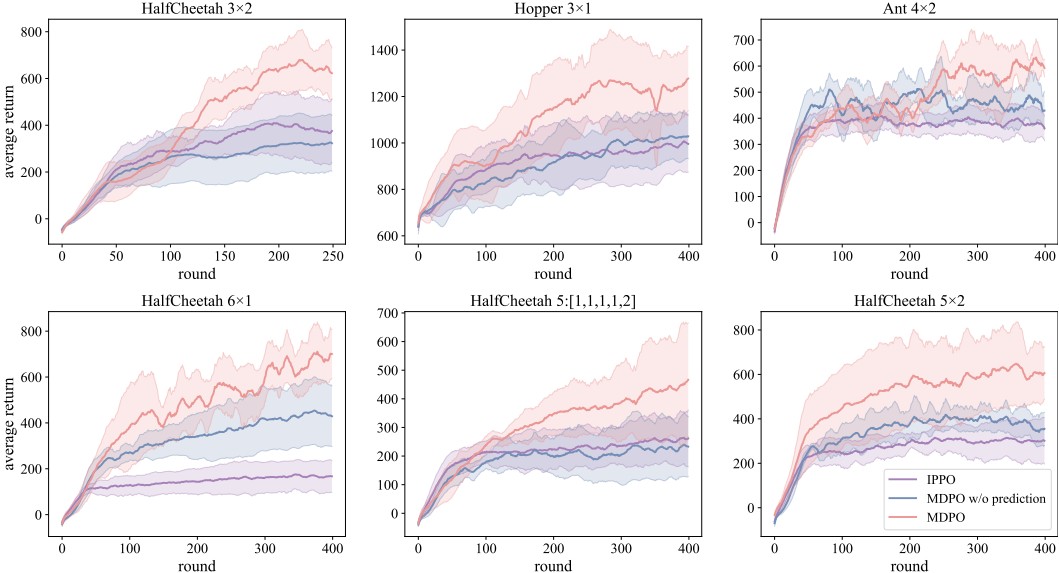

Figure 4: Learning curves of MDPO, MDPO w/o prediction, and IPPO in six multi-agent MuJoCo tasks. Each round is 4000 environment steps for 4-agent Ant and 2000 for other tasks.

matter to form a regular polygon in Regular Polygon Control and are hard to learn accurately. Thus, MDPO performs well in Cooperative Navigation since the very early learning stage, while it does not perform well in Regular Polygon Control before 600 rounds. Although a more accurate model is required in Regular Polygon Control, MDPO still converges to better performance. And this indicates a progressive pattern in prediction also works when prediction is hard to be fairly accurate.

### 4.3 MULTI-AGENT MUJOCO

Multi-agent MuJoCo is a continuous multi-agent robotic control environment, based on OpenAI's Mujoco Gym environments. In a multi-agent MuJoCo task, each agent controls several joints of the robotic to move forward, where both the observation space and action space are continuous. We choose 3-agent Hopper, 4-agent Ant, and 4 versions of HalfCheetah with different agent numbers or joint allocation for performance comparison. Details of joint allocation are given in Appendix C.

As illustrated in Figure 4, MDPO consistently performs better in these tasks with different difficulties and various agent numbers. Compared with MPE, agents in multi-agent MuJoCo have deeper impacts on each other due to the interaction between adjacent joints. Consequently, the transitions of each agent are closely related to the policies of other agents. Thus, non-stationarity caused by policy updates of other agents is severer in these tasks, resulting in IPPO struggling and converging to low performance. Moreover, note that MDPO w/o prediction performs almost the same as IPPO or even worse in some tasks. The poor performance of MDPO w/o prediction is a consequence of a larger $\epsilon_\psi$ caused by strongly associated agents in these tasks. Thus, latent variable prediction is necessary in these tasks with closely associated agents.

## 5 CONCLUSION

In this paper, we propose model-based decentralized policy optimization (MDPO). By introducing a latent variable into the environment model, we theoretically show the model helps to generate experiences with more stationary observation visitation frequency and benefits decentralized policy optimization. Furthermore, We theoretically analyze that the return bound for monotonic policy improvement is controllable by the prediction error of the latent variable function. Consequently, we propose a latent variable prediction method to constrain the prediction error. We examine all the theories and designs via experiments on a set of cooperative multi-agent tasks. Results verify our theoretical results and show MDPO indeed obtains superior performance over model-free decentralized policy optimization.

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

# A PROOFS

## A.1 OBSERVATION VISITATION FREQUENCY DIVERGENCE

In this section, we provide proofs for the upper bound of observation visitation frequency divergence.

**Lemma 1.** *Given two pairs of policy and latent variable function, $(\pi_1, \psi_1)$ and $(\pi_2, \psi_2)$. $\forall o \in \mathcal{O}$, it holds that*

$$\sum_{a,z} |\pi_1(a|o)\psi_1(z|o) - \pi_2(a|o)\psi_2(z|o)| \leq |\mathcal{A}| \cdot \|\pi_1 - \pi_2\| + |\mathcal{Z}| \cdot \|\psi_1 - \psi_2\|,$$

*where $\|\pi_1 - \pi_2\| \triangleq \max\limits_{a,o}|\pi_1(a|o) - \pi_2(a|o)|$, $\|\psi_1 - \psi_2\| \triangleq \max\limits_{z,o}|\psi_1(z|o) - \psi_2(z|o)|$.*

*Proof.*

$$
\begin{aligned}
\sum_{a,z} |\pi_1(a|o)\psi_1(z|o) - \pi_2(a|o)\psi_2(z|o)| \leq & \sum_{a,z} |\pi_1(a|o)\psi_1(z|o) - \pi_1(a|o)\psi_2(z|o)| \\
& + \sum_{a,z} |\pi_1(a|o)\psi_2(z|o) - \pi_2(a|o)\psi_2(z|o)| \\
= & \sum_a \pi_1(a|o) \sum_z |\psi_1(z|o) - \psi_2(z|o)| \\
& + \sum_z \psi_2(z|o) \sum_a |\pi_1(a|o) - \pi_2(a|o)| \\
\leq & |\mathcal{A}| \cdot \|\pi_1 - \pi_2\| + |\mathcal{Z}| \cdot \|\psi_1 - \psi_2\|.
\end{aligned}
$$

$\square$

**Lemma 2** (Timestep observation visitation frequency recursion). *Given two pairs of policy and latent variable function $(\pi_1, \psi_1)$ and $(\pi_2, \psi_2)$, we define :*

$$\Delta\rho_t^{(\pi_1,\psi_1),(\pi_2,\psi_2)}(o) \triangleq \rho_t^{\pi_1,\psi_1}(o) - \rho_t^{\pi_2,\psi_2}(o),$$

$$\left\|\Delta\rho_t^{(\pi_1,\psi_1),(\pi_2,\psi_2)}\right\| \triangleq \max_o \left|\Delta\rho_t^{(\pi_1,\psi_1),(\pi_2,\psi_2)}(o)\right|.$$

*It holds that*

$$\left|\Delta\rho_{t+1}^{(\pi_1,\psi_1),(\pi_2,\psi_2)}(o')\right| \leq |\mathcal{A}| \cdot \|\pi_1 - \pi_2\| + |\mathcal{Z}| \cdot \|\psi_1 - \psi_2\| + |\mathcal{O}| \cdot \left\|\Delta\rho_t^{(\pi_1,\psi_1),(\pi_2,\psi_2)}\right\|.$$

*Proof.* For observation visitation frequency at timestep $t+1$, there is a recurrence relation:

$$\rho_{t+1}^{\pi,\psi}(o') = \sum_o \rho_t^{\pi,\psi}(o) \sum_{a,z} P(o'|a,o,z)\pi(a|o)\psi(z|o)$$

Thus, the divergence of observation visitation frequency at timestep $t+1$ can be processed correspondingly:

$$
\begin{aligned}
\Delta\rho_{t+1}^{(\pi_1,\psi_1),(\pi_2,\psi_2)}(o') = & \rho_{t+1}^{\pi_1,\psi_1}(o') - \rho_{t+1}^{\pi_2,\psi_2}(o') \\
= & \sum_o \left(\rho_t^{\pi_1,\psi_1}(o) \sum_{a,z}(P(o'|a,o,z)\pi_1(a|s)\psi_1(z|o))\right) \\
& - \sum_o \left(\rho_t^{\pi_2,\psi_2}(o) \sum_{a,z}(P(o'|a,o,z)\pi_2(a|s)\psi_2(z|o))\right) \\
= & \sum_o \left(\rho_t^{\pi_1,\psi_1}(o) \sum_{a,z}(P(o'|a,o,z)(\pi_1(a|o)\psi_1(z|o) - \pi_2(a|o)\psi_2(z|o)))\right) \\
& + \sum_o \left(\Delta\rho_t^{(\pi_1,\psi_1),(\pi_2,\psi_2)}(o) \sum_{a,z}(P(o'|a,o,z)\pi_2(a|o)\psi_2(z|o))\right)
\end{aligned}
$$

Using Lemma 1, we can bound the divergence of observation state frequency at timestep $t + 1$:

$$
\begin{aligned}
\left| \Delta \rho_{t+1}^{(\pi_1, \psi_1),(\pi_2, \psi_2)} (o') \right| &\leq \sum_o \left( \rho_t^{\pi_1, \psi_1} (o) \sum_{a,z} \left( P\left(o'|a, o, z\right) \left| \pi_1 (a|o) \psi_1 (z|o) - \pi_2 (a|o) \psi_2 (z|o) \right| \right) \right) \\
&\quad + \sum_o \left( \left| \Delta \rho_t^{(\pi_1, \psi_1),(\pi_2, \psi_2)} (o) \right| \sum_{a,z} \left( P\left(o'|a, o, z\right) \pi_2 (a|o) \psi_2 (z|o) \right) \right) \\
&\leq \sum_o \left( \rho_t^{\pi_1, \psi_1} (o) \left( |\mathcal{A}| \, \|\pi_1 - \pi_2\| + |\mathcal{Z}| \, \|\psi_1 - \psi_2\| \right) \right) \\
&\quad + \left\| \Delta \rho_t^{(\pi_1, \psi_1),(\pi_2, \psi_2)} \right\| \sum_{o,a,z} \left( P\left(o'|a, o, z\right) \pi_2 (a|o) \psi_2 (z|o) \right) \\
&\leq |\mathcal{A}| \cdot \|\pi_1 - \pi_2\| + |\mathcal{Z}| \cdot \|\psi_1 - \psi_2\| + |\mathcal{O}| \cdot \left\| \Delta \rho_t^{(\pi_1, \psi_1),(\pi_2, \psi_2)} \right\|
\end{aligned}
$$

$\square$

**Lemma 3** (discounted observation visitation frequency divergence bound). *Given two pairs of policy and latent variable function $(\pi_1, \psi_1)$ and $(\pi_2, \psi_2)$, with the same distribution of initial observation $\rho_0 (o)$, it holds that*

$$
\left\| \Delta \rho^{(\pi_1, \psi_1),(\pi_2, \psi_2)} \right\|_1 \leq C \left( \|\pi_1 - \pi_2\| + \|\psi_1 - \psi_2\| \right),
$$

*where $C$ is a certain constant.*

*Proof.* We transform it to the cumulative form of the timestep, and scale it using Lemma 2:

$$
\begin{aligned}
\left| \Delta \rho^{(\pi_1, \psi_1),(\pi_2, \psi_2)} (o) \right| &= \left| \sum_{t=0}^{\infty} \gamma^t \Delta \rho_t^{(\pi_1, \psi_1),(\pi_2, \psi_2)} (o) \right| \\
&\leq \sum_{t=1}^{T-1} \gamma^t \left| \Delta \rho_t^{(\pi_1, \psi_1),(\pi_2, \psi_2)} (o) \right| + \gamma^T \sum_{t=T}^{\infty} \gamma^{t-T} \left| \Delta \rho_t^{(\pi_1, \psi_1),(\pi_2, \psi_2)} (o) \right| \\
&\leq \sum_{t=1}^{T-1} \gamma^t \left( |\mathcal{A}| \, \|\pi_1 - \pi_2\| + |\mathcal{Z}| \, \|\psi_1 - \psi_2\| \right) \\
&\quad + \gamma |\mathcal{O}| \sum_{t=1}^{T-2} \gamma^t \left\| \Delta \rho_t^{(\pi_1, \psi_1),(\pi_2, \psi_2)} \right\| + \frac{2 \gamma^T}{1 - \gamma} \\
&\leq \left( \sum_{t=1}^{T-1} \gamma^t \sum_{k=0}^{T-1-t} (\gamma |\mathcal{O}|)^k \right) \left( |\mathcal{A}| \, \|\pi_1 - \pi_2\| + |\mathcal{Z}| \, \|\psi_1 - \psi_2\| \right) + \frac{2 \gamma^T}{1 - \gamma}.
\end{aligned}
$$

Thus, we get bound discounted observation visitation frequency divergence:

$$
\left\| \Delta \rho^{(\pi_1, \psi_1),(\pi_2, \psi_2)} \right\|_{\infty} \leq C_1 \left( \|\pi_1 - \pi_2\| + \|\psi_1 - \psi_2\| \right),
$$

$$
\left\| \Delta \rho^{(\pi_1, \psi_1),(\pi_2, \psi_2)} \right\|_1 \leq |\mathcal{O}| \cdot \left\| \Delta \rho^{(\pi_1, \psi_1),(\pi_2, \psi_2)} \right\|_{\infty} \leq C_2 \left( \|\pi_1 - \pi_2\| + \|\psi_1 - \psi_2\| \right).
$$

$\square$

## A.2 LATENT VARIABLE FUNCTION DIVERGENCE

In this section, we provide proof for divergence comparison between latent variable function of policy rollout and learned latent variable function in the model.

**Lemma 4** (Latent variable function divergence comparison). *Assume* $\psi_\omega^n = (1-\alpha)\psi_\omega^{n-1} + \alpha\psi^n$, *and initially* $\psi_\omega^1 = \psi^1$, *where $n$ is the $n$th policy rollout. Then,*

$$\mathcal{E}_\psi > \mathcal{E}_{\psi_\omega},$$

*where* $\mathcal{E}_\psi \triangleq \max_n \|\psi^n - \psi^{n-1}\|$ *and* $\mathcal{E}_{\psi_\omega} \triangleq \max_n \|\psi_\omega^n - \psi_\omega^{n-1}\|$.

*Proof.* Firstly, we can construct such a recursive inequality:

$$\|\psi^n - \psi_\omega^{n-1}\| = \|\psi^n - (1-\alpha)\psi_\omega^{n-2} - \alpha\psi^{n-1}\| \leq \|\psi^n - \psi^{n-1}\| + (1-\alpha)\|\psi^{n-1} - \psi_\omega^{n-2}\|.$$

Thus, we can expand it recursively:

$$\begin{aligned}
\|\psi^n - \psi_\omega^{n-1}\| &\leq \|\psi^n - \psi^{n-1}\| + (1-\alpha)\|\psi^{n-1} - \psi_\omega^{n-2}\| \\
&\leq \|\psi^n - \psi^{n-1}\| + (1-\alpha)\|\psi^{n-1} - \psi_\omega^{n-2}\| + \cdots + (1-\alpha)^{n-1}\|\psi^1 - \psi_\omega^0\| \\
&= \|\psi^n - \psi^{n-1}\| + (1-\alpha)\|\psi^{n-1} - \psi_\omega^{n-2}\| + \cdots + (1-\alpha)^{n-1}\|\psi^1 - \psi^0\| \\
&\leq \mathcal{E}_\psi\left(1 + (1-\alpha) + \cdots + (1-\alpha)^{n-1}\right) \\
&< \frac{\mathcal{E}_\psi}{\alpha}.
\end{aligned}$$

Using this inequality, we can zoom $\|\psi_\omega^n - \psi_\omega^{n-1}\|$:

$$\|\psi_\omega^n - \psi_\omega^{n-1}\| = \|(1-\alpha)\psi_\omega^{n-1} + \alpha\psi^n - \psi_\omega^{n-1}\| = \alpha\|\psi^n - \psi_\omega^{n-1}\| z < \mathcal{E}_\psi.$$

Thus, $\mathcal{E}_{\psi_\omega} = \max_n \|\psi_\omega^n - \psi_\omega^{n-1}\| < \mathcal{E}_\psi$.

$\square$

Now we combine Lemma 3 and 4 to prove Theorem 1.

**Theorem 1** (Latent variable model benefits). *Define* $\Delta\rho^n(o) \triangleq \rho^{\pi^n,\psi^n}(o) - \rho^{\pi^{n-1},\psi^{n-1}}(o)$, *and denote* $\|\Delta\rho^n\| \triangleq \max_o |\rho^n(o) - \psi^{n-1}(o)|$, *similarly for* $\|\Delta\pi^n\|$ *and* $\|\Delta\psi^n\|$. *It holds that,*

$$\|\Delta\rho^n\| \leq C(\mathcal{E}_\pi + \mathcal{E}_\psi),$$

*where* $\mathcal{E}_\pi \triangleq \max_n \|\Delta\pi^n\|$, $\mathcal{E}_\psi \triangleq \max_n \|\Delta\psi^n\|$ *and $C$ is a constant. Assume* $\psi_\omega^n = (1-\alpha)\psi_\omega^{n-1} + \alpha\psi^n$, *and initially* $\psi_\omega^0 = \psi^0$.[3] *It holds that* $\mathcal{E}_\psi > \mathcal{E}_{\psi_\omega}$ *and the bound above is lower when substituting $\psi$ with $\psi_\omega$.*

*Proof.* Using Lemma 3, we can scale $\|\Delta\rho^n\|$,

$$\|\Delta\rho^n\| \leq C(\|\Delta\pi^n\| + \|\Delta\psi^n\|) \leq C(\mathcal{E}_\pi + \mathcal{E}_\psi)$$

Using Lemma 4, $\mathcal{E}_\psi > \mathcal{E}_{\psi_\omega}$. Thus, the bound above is lower for $\mathcal{E}_{\psi_\omega}$.

$\square$

### A.3 LEMMAS FOR RETURN BOUND ANALYSIS

In this section, we prove several lemmas as preparations for return bound analysis.

**Lemma 5** (TVD bound of joint distribution). *Consider two joint distributions of $n + 1$ variables like this:*

$$P_1(x, y_1, y_2, \cdots, y_n) = P_1(x) \cdot \prod_{i=1}^n P_1(y_i|x)$$

$$P_2(x, y_1, y_2, \cdots, y_n) = P_2(x) \cdot \prod_{i=1}^n P_2(y_i|x)$$

---

[3] Since $\psi$ is varying and $\psi_\omega$ is continuously updated using the experiences from several recent policy rollouts, we use the form of soft-update for the relation between $\psi$ and $\psi_\omega$.

*We can bound the total variation distance of the joint distributions as:*

$$D_{TV}\left(P_1\left(x,y_1,\cdots,y_n\right)\|P_2\left(x,y_1,\cdots,y_n\right)\right) \leq D_{TV}\left(P_1\left(x\right)\|P_2\left(x\right)\right) + \sum_{i=1}^{n}\max_{x} D_{TV}\left(P_1\left(y_i|x\right)\|P_2\left(y_i|x\right)\right).$$

*Proof.* We start the proof from a basis case when $n = 1$:

$$
\begin{aligned}
D_{TV}\left(P_1\|P_2\right) =& \frac{1}{2}\sum_{x,y}|P_1\left(x,y\right) - P_2\left(x,y\right)| \\
\leq& \frac{1}{2}\sum_{x,y}|P_1\left(x\right)P_1\left(y|x\right) - P_2\left(x\right)P_1\left(y|x\right)| + |P_2\left(x\right)P_1\left(y|x\right) - P_2\left(x\right)P_2\left(y|x\right)| \\
=& \frac{1}{2}\sum_{x,y}|P_1\left(x\right) - P_2\left(x\right)| \cdot P_1\left(y|x\right) + P_2\left(x\right) \cdot |P_1\left(y|x\right) - P_2\left(y|x\right)| \\
=& \frac{1}{2}\sum_{x}|P_1\left(x\right) - P_2\left(x\right)| + \sum_{x}P_2\left(x\right)D_{TV}\left(P_1\left(y|x\right)\|P_2\left(y|x\right)\right) \\
\leq& D_{TV}\left(P_1\left(x\right)\|P_2\left(x\right)\right) + \max_{x}D_{TV}\left(P_1\left(y|x\right)\|P_2\left(y|x\right)\right)
\end{aligned}
$$

Similarly, we can prove the case of multi-variables:

$$
\begin{aligned}
D_{TV}\left(P_1\|P_2\right) \leq& D_{TV}\left(P_1\left(x\right)\|P_2\left(x\right)\right) + \max_{x}D_{TV}\left(P_1\left(y_1,\cdots,y_n|x\right)\|P_2\left(y_1,\cdots,y_n|x\right)\right) \\
\leq& D_{TV}\left(P_1\left(x\right)\|P_2\left(x\right)\right) + \max_{x}\Big[D_{TV}\left(P_1\left(y_1|x\right)\|P_2\left(y_1|x\right)\right) \\
& \qquad\qquad + \max_{x}D_{TV}\left(P_1\left(y_2,\cdots,y_n|x\right)\|P_2\left(y_2,\cdots,y_n|x\right)\right)\Big] \\
=& D_{TV}\left(P_1\left(x\right)\|P_2\left(x\right)\right) + \max_{x}D_{TV}\left(P_1\left(y_1|x\right)\|P_2\left(y_1|x\right)\right) \\
& \qquad\qquad + \max_{x}D_{TV}\left(P_1\left(y_2,\cdots,y_n|x\right)\|P_2\left(y_2,\cdots,y_n|x\right)\right) \\
\leq& D_{TV}\left(P_1\left(x\right)\|P_2\left(x\right)\right) + \sum_{i=1}^{n}\max_{x}D_{TV}\left(P_1\left(y_i|x\right)\|P_2\left(y_i|x\right)\right)
\end{aligned}
$$

$\square$

Before proving following lemmas, we clarify the premise our discuss in this section is based on. In a Dec-POMDP, denote the co-occurrence probability of tuple $(o, a, z)$ at timestep $t$ as $P^t(o, a, z) \triangleq P(o_t = o, a_t = a, z_t = z)$.

Consider two Dec-POMDPs different merely in transition function and reward function, $G_1, G_2$. $P_1$ represents the probability in $G_1$ while $P_2$ for $G_2$. Different policies and latent variable functions, $(\pi_1, \psi_1)$ and $(\pi_2, \psi_2)$, are used to rollout respectively in $G_1$ and $G_2$, We denote several bound between them:

reward bound: $r_{\max} \triangleq \max_{o,a,z}\max\{R_1(o, a, z), R_2(o, a, z)\}$;

policy bound: $\epsilon_\pi \triangleq \max_{o}D_{TV}\left(\pi_1\|\pi_2\right)$;

latent variable function bound: $\epsilon_\psi \triangleq \max_{o}D_{TV}\left(\psi_1\|\psi_2\right)$;

transition function bound: $\epsilon_m \triangleq \max_{t}\mathbb{E}_{o,a,z\sim P_2^{t-1}}D_{TV}\left(P_1\left(o_t|o, a, z\right)\|P_2\left(o_t|o, a, z\right)\right)$.

Additionally, consider the branched model rollout mentioned in Section 3. Policy, latent variable function, transition function, and reward function vary before and after the model rollout branch. We

denote these functions via superscripts 'Pre' for function before branch and 'Post' for functions after branch. Correspondingly, when discussing branched model rollout, we extend the bounds above:

reward bound: $r_{\max} \triangleq \max\limits_{o,a,z} \max\{R_1^{Pre}(o,a,z), R_1^{Post}(o,a,z), R_2^{Pre}(o,a,z), R_2^{Post}(o,a,z)\}$;

policy bound: $\epsilon_\pi^{Pre} \triangleq \max\limits_{o} D_{TV}\left(\pi_1^{Pre} \| \pi_2^{Pre}\right), \epsilon_\pi^{Post} \triangleq \max\limits_{o} D_{TV}\left(\pi_1^{Post} \| \pi_2^{Post}\right)$;

latent variable function bound: $\epsilon_\psi^{Pre} \triangleq \max\limits_{o} D_{TV}\left(\psi_1^{Pre} \| \psi_2^{Pre}\right), \epsilon_\psi^{Post} \triangleq \max\limits_{o} D_{TV}\left(\psi_1^{Post} \| \psi_2^{Post}\right)$;

transition function bound: $\epsilon_m^{Pre} \triangleq \max\limits_{t} \mathbb{E}_{o,a,z \sim P_2^{t-1,Pre}} D_{TV}\left(P_1^{Pre}(o_t|o,a,z) \| P_2^{Pre}(o_t|o,a,z)\right)$,

$$\epsilon_m^{Post} \triangleq \max\limits_{t} \mathbb{E}_{o,a,z \sim P_2^{t-1,Post}} D_{TV}\left(P_1^{Post}(o_t|o,a,z) \| P_2^{Post}(o_t|o,a,z)\right).$$

**Lemma 6** (Observation distributions TVD bound). *The total variation distance of observation distributions at timestep $t$, $P_1(o_t)$ and $P_2(o_t)$, can be bounded as below:*

$$D_{TV}\left(P_1(o_t) \| P_2(o_t)\right) \le t\left(\epsilon_\pi + \epsilon_\psi + \epsilon_m\right)$$

.

*Proof.*

$$D_{TV}\left(P_1(o_t) \| P_2(o_t)\right) = \frac{1}{2}\sum_{o_t}|P_1(o_t) - P_2(o_t)|$$

$$= \frac{1}{2}\sum_{o_t}\left|\sum_{o,a,z} P_1^{t-1}(o,a,z) P_1(o_t|o,a,z) - P_2^{t-1}(o,a,z) P_2(o_t|o,a,z)\right|$$

$$\le \frac{1}{2}\sum_{o_t}\left(\sum_{o,a,z}\left|P_1^{t-1}(o,a,z) - P_2^{t-1}(o,a,z)\right| P_1(o_t|o,a,z)\right.$$

$$\left. + \sum_{o,a,z} P_2^{t-1}(o,a,z)\left|P_1(o_t|o,a,z) - P_2(o_t|o,a,z)\right|\right)$$

$$= \frac{1}{2}\sum_{o,a,z}\left|P_1^{t-1}(o,a,z) - P_2^{t-1}(o,a,z)\right|\sum_{o_t} P_1(o_t|o,a,z)$$

$$+ \mathbb{E}_{o,a,z \sim P_2^{t-1}} D_{TV}\left(P_1(o_t|o,a,z) \| P_2(o_t|o,a,z)\right)$$

$$= D_{TV}\left(P_1^{t-1}(o,a,z) \| P_2^{t-1}(o,a,z)\right) + \epsilon_m$$

According to Lemma 5,

$$D_{TV}\left(P_1^{t-1}(o,a,z) \| P_2^{t-1}(o,a,z)\right) \le D_{TV}\left(P_1(o_{t-1}) \| P_2(o_{t-1})\right)$$

$$+ \max\limits_{o} D_{TV}\left(\pi_1 \| \pi_2\right) + \max\limits_{o} D_{TV}\left(\psi_1 \| \psi_2\right)$$

$$= D_{TV}\left(P_1(o_{t-1}) \| P_2(o_{t-1})\right) + \epsilon_\pi + \epsilon_\psi$$

Thus,

$$D_{TV}\left(P_1(o_t) \| P_2(o_t)\right) \le D_{TV}\left(P_1(o_{t-1}) \| P_2(o_{t-1})\right) + \epsilon_\pi + \epsilon_\psi + \epsilon_m$$

$$\le D_{TV}\left(P_1(o_0) \| P_2(o_0)\right) + t\left(\epsilon_\pi + \epsilon_\psi + \epsilon_m\right)$$

$$= t\left(\epsilon_\pi + \epsilon_\psi + \epsilon_m\right)$$

$\square$

**Lemma 7** (Rollout return bound). *The gap between rollout returns in $G_1$ with $(\pi_1, \psi_1)$ and $G_2$ with $(\pi_2, \psi_2)$ is bounded as:*

$$|\eta_1(\pi_1, \psi_1) - \eta_2(\pi_2, \psi_2)| \le \frac{2r_{\max}}{1-\gamma}\left(\frac{\gamma(\epsilon_\pi + \epsilon_\psi + \epsilon_m)}{1-\gamma} + \epsilon_\pi + \epsilon_\psi\right)$$

*Proof.*

$$|\eta_1\left(\pi_1, \psi_1\right) - \eta_2\left(\pi_2, \psi_2\right)| = \left| \sum_{o,a,z} R\left(o, a, z\right) \sum_t \gamma^t \left(P_1^t\left(o, a, z\right) - P_2^t\left(o, a, z\right)\right) \right|$$

$$\leq 2r_{\max} \sum_t \gamma^t D_{TV}\left(P_1^t\left(o, a, z\right) \| P_2^t\left(o, a, z\right)\right)$$

Using Lemma 5 and 6,

$$D_{TV}\left(P_1^t\left(o, a, z\right) \| P_2^t\left(o, a, z\right)\right) \leq D_{TV}\left(P_1\left(o_t\right) \| P_2\left(o_t\right)\right) + \max_o D_{TV}\left(\pi_1 \| \pi_2\right) + \max_o D_{TV}\left(\psi_1 \| \psi_2\right)$$

$$\leq t\left(\epsilon_\pi + \epsilon_\psi + \epsilon_m\right) + \epsilon_\pi + \epsilon_\psi$$

Thus,

$$|\eta_1\left(\pi_1, \psi_1\right) - \eta_2\left(\pi_2, \psi_2\right)| \leq \frac{2r_{\max}}{1-\gamma} \left( \frac{\gamma\left(\epsilon_\pi + \epsilon_\psi + \epsilon_m\right)}{1-\gamma} + \epsilon_\pi + \epsilon_\psi \right)$$

$\square$

**Lemma 8** (Branched rollout return bound). *In $k$-step branched rollout with $h$-step length before the branch taking into consideration, denote the gap between branched rollout returns in $G_1$ with $\left(\left(\pi_1^{Pre}, \pi_1^{Post}\right), \left(\psi_1^{Pre}, \psi_1^{Post}\right)\right)$ and $G_2$ with $\left(\left(\pi_2^{Pre}, \pi_2^{Post}\right), \left(\psi_2^{Pre}, \psi_2^{Post}\right)\right)$ as:*

$$|\eta_1 - \eta_2| \triangleq \left| \eta_1^{branch}\left(\left(\pi_1^{Pre}, \pi_1^{Post}\right), \left(\psi_1^{Pre}, \psi_1^{Post}\right)\right) - \eta_2^{branch}\left(\left(\pi_2^{Pre}, \pi_2^{Post}\right), \left(\psi_2^{Pre}, \psi_2^{Post}\right)\right) \right|$$

*which is bounded as:*

$$|\eta_1 - \eta_2| \leq \frac{2r_{\max}}{1-\gamma} \Bigg[ \left( h + \frac{\gamma^{h+k+1}}{(1-\gamma)} \right) \left(\epsilon_\pi^{Pre} + \epsilon_\psi^{Pre} + \epsilon_m^{Pre}\right) + \left(\epsilon_\pi^{Pre} + \epsilon_\psi^{Pre}\right)$$
$$+ \gamma^{h+1} \left( k \left(\epsilon_\pi^{Post} + \epsilon_\psi^{Post} + \epsilon_m^{Post}\right) + \epsilon_\pi^{Post} + \epsilon_\psi^{Post}\right) \Bigg]$$

*Proof.* According to Lemma 6,

$$D_{TV}\left(P_1\left(o_t\right) \| P_2\left(o_t\right)\right) \leq D_{TV}\left(P_1\left(o_{t-1}\right) \| P_2\left(o_{t-1}\right)\right) + \epsilon_\pi + \epsilon_\psi + \epsilon_m,$$

which stays in branched rollout case.

We can discuss $\delta_t \triangleq D_{TV}\left(P_1^t\left(o, a, z\right) \| P_2^t\left(o, a, z\right)\right)$ with different $t$ value:

when $t \leq h$,

$$\delta_t \leq t\left(\epsilon_\pi^{Pre} + \epsilon_\psi^{Pre} + \epsilon_\omega^{Pre}\right) + \epsilon_\pi^{Pre} + \epsilon_\psi^{Pre}$$

when $h < t \leq h + k$,

$$\delta_t \leq h\left(\epsilon_\pi^{Pre} + \epsilon_\psi^{Pre} + \epsilon_m^{Pre}\right) + (t-h)\left(\epsilon_\pi^{Post} + \epsilon_\psi^{Post} + \epsilon_m^{Post}\right) + \epsilon_\pi^{Post} + \epsilon_\psi^{Post}$$

when $t > h + k$,

$$\delta_t \leq h\left(\epsilon_\pi^{Pre} + \epsilon_\psi^{Pre} + \epsilon_m^{Pre}\right) + k\left(\epsilon_\pi^{Post} + \epsilon_\psi^{Post} + \epsilon_m^{Post}\right) + \epsilon_\pi^{Post} + \epsilon_\psi^{Post}$$
$$+ (t-h-k)\left(\epsilon_\pi^{Pre} + \epsilon_\psi^{Pre} + \epsilon_m^{Pre}\right) + \epsilon_\pi^{Pre} + \epsilon_\psi^{Pre}$$
$$= (t-k)\left(\epsilon_\pi^{Pre} + \epsilon_\psi^{Pre} + \epsilon_m^{Pre}\right) + \epsilon_\pi^{Pre} + \epsilon_\psi^{Pre} + k\left(\epsilon_\pi^{Post} + \epsilon_\psi^{Post} + \epsilon_m^{Post}\right) + \epsilon_\pi^{Post} + \epsilon_\psi^{Post}$$

Using the inequalities above, we can write:

$$\delta \triangleq \sum_t \gamma^t D_{TV}\left(P_1^t(o,a,z) \| P_2^t(o,a,z)\right)$$

$$\leq \sum_{t=0}^{h}\left[\gamma^t\left(t\left(\epsilon_\pi^{Pre}+\epsilon_\psi^{Pre}+\epsilon_m^{Pre}\right)+\epsilon_\pi^{Pre}+\epsilon_\psi^{Pre}\right)\right]$$

$$+\sum_{t=h+1}^{h+k}\left[\gamma^t\left(h\left(\epsilon_\pi^{Pre}+\epsilon_\psi^{Pre}+\epsilon_m^{Pre}\right)+\epsilon_\pi^{Post}+\epsilon_\psi^{Post}+(t-h)\left(\epsilon_\pi^{Post}+\epsilon_\psi^{Post}+\epsilon_m^{Post}\right)\right)\right]$$

$$+\sum_{t=h+k+1}^{\infty}\left[\gamma^t\left((t-k)\left(\epsilon_\pi^{Pre}+\epsilon_\psi^{Pre}+\epsilon_m^{Pre}\right)+\epsilon_\pi^{Pre}+\epsilon_\psi^{Pre}\right.\right.$$

$$\left.\left.+k\left(\epsilon_\pi^{Post}+\epsilon_\psi^{Post}+\epsilon_m^{Post}\right)+\epsilon_\pi^{Post}+\epsilon_\psi^{Post}\right)\right]$$

$$\leq \sum_{t=0}^{h}\left[\gamma^t\left(h\left(\epsilon_\pi^{Pre}+\epsilon_\psi^{Pre}+\epsilon_m^{Pre}\right)+\epsilon_\pi^{Pre}+\epsilon_\psi^{Pre}\right)\right]$$

$$+\sum_{t=h+1}^{h+k}\left[\gamma^t\left(h\left(\epsilon_\pi^{Pre}+\epsilon_\psi^{Pre}+\epsilon_m^{Pre}\right)+\epsilon_\pi^{Pre}+\epsilon_\psi^{Pre}\right.\right.$$

$$\left.\left.+k\left(\epsilon_\pi^{Post}+\epsilon_\psi^{Post}+\epsilon_m^{Post}\right)+\epsilon_\pi^{Post}+\epsilon_\psi^{Post}\right)\right]$$

$$+\sum_{t=h+k+1}^{\infty}\left[\gamma^t\left((t-k)\left(\epsilon_\pi^{Pre}+\epsilon_\psi^{Pre}+\epsilon_m^{Pre}\right)+\epsilon_\pi^{Pre}+\epsilon_\psi^{Pre}\right.\right.$$

$$\left.\left.+k\left(\epsilon_\pi^{Post}+\epsilon_\psi^{Post}+\epsilon_m^{Post}\right)+\epsilon_\pi^{Post}+\epsilon_\psi^{Post}\right)\right]$$

$$\leq\left(\sum_{t=0}^{\infty}\gamma^t h+\gamma^{h+k}\sum_{t=0}^{\infty}\gamma^t t\right)\left(\epsilon_\pi^{Pre}+\epsilon_\psi^{Pre}+\epsilon_m^{Pre}\right)+\left(\sum_{t=0}^{\infty}\gamma^t\right)\left(\epsilon_\pi^{Pre}+\epsilon_\psi^{Pre}\right)$$

$$+\left(\sum_{t=h+1}^{\infty}\gamma^t\right)\left(k\left(\epsilon_\pi^{Post}+\epsilon_\psi^{Post}+\epsilon_m^{Post}\right)+\epsilon_\pi^{Post}+\epsilon_\psi^{Post}\right)$$

$$=\left(\frac{h}{1-\gamma}+\frac{\gamma^{h+k+1}}{(1-\gamma)^2}\right)\left(\epsilon_\pi^{Pre}+\epsilon_\psi^{Pre}+\epsilon_m^{Pre}\right)+\frac{1}{1-\gamma}\left(\epsilon_\pi^{Pre}+\epsilon_\psi^{Pre}\right)$$

$$+\frac{\gamma^{h+1}}{1-\gamma}\left(k\left(\epsilon_\pi^{Post}+\epsilon_\psi^{Post}+\epsilon_m^{Post}\right)+\epsilon_\pi^{Post}+\epsilon_\psi^{Post}\right)$$

Thus,

$$|\eta_1-\eta_2| \leq 2r_{\max}\delta$$

$$\leq\frac{2r_{\max}}{1-\gamma}\left[\left(h+\frac{\gamma^{h+k+1}}{(1-\gamma)}\right)\left(\epsilon_\pi^{Pre}+\epsilon_\psi^{Pre}+\epsilon_m^{Pre}\right)+\left(\epsilon_\pi^{Pre}+\epsilon_\psi^{Pre}\right)\right.$$

$$\left.+\gamma^{h+1}\left(k\left(\epsilon_\pi^{Post}+\epsilon_\psi^{Post}+\epsilon_m^{Post}\right)+\epsilon_\pi^{Post}+\epsilon_\psi^{Post}\right)\right]$$

$\square$

### A.4 PROOF OF RETURN BOUND

In this section, we provide proofs of return bound in different cases.

**Theorem 2** (Rollout return bound for decentralized model) **.** *The gap between return in $n + 1$th policy rollout and return in model rollout with $n$th learned model $\eta^{model}(\pi, \psi_\omega^n)$ is bounded as:*

$$\left| \eta\left(\pi, \psi^{n+1}\right) - \eta^{model}\left(\pi, \psi_\omega^n\right) \right| \leq \underbrace{\frac{2r_{\max}}{(1 - \gamma)^2}\left(\gamma\epsilon_\theta + 2\epsilon_\pi + 2\epsilon_\omega + \epsilon_\psi\right)}_{C(\epsilon_\theta, \epsilon_\pi, \epsilon_\omega, \epsilon_\psi)}.$$

*Proof.*

$$\left| \eta\left(\pi, \psi^{n+1}\right) - \eta^{model}\left(\pi, \psi_\omega^n\right) \right| \leqslant \underbrace{\left| \eta\left(\pi, \psi^{n+1}\right) - \eta\left(\pi_D^n, \psi^n\right) \right|}_{L_1} + \underbrace{\left| \eta\left(\pi_D^n, \psi^n\right) - \eta^{model}\left(\pi, \psi_\omega^n\right) \right|}_{L_2}$$

Apply Lemma 7 to $L_1$ and $L_2$:

$$L_1 \leq \frac{2r_{\max}}{1 - \gamma}\left(\frac{\gamma\left(\epsilon_\pi + \epsilon_\psi\right)}{1 - \gamma} + \epsilon_\pi + \epsilon_\psi\right),$$

$$L_2 \leq \frac{2r_{\max}}{1 - \gamma}\left(\frac{\gamma\left(\epsilon_\pi + \epsilon_\theta + \epsilon_\omega\right)}{1 - \gamma} + \epsilon_\pi + \epsilon_\omega\right).$$

Thus,

$$\left| \eta\left(\pi, \psi^{n+1}\right) - \eta^{model}\left(\pi, \psi_\omega^n\right) \right| \leq \frac{2r_{\max}}{(1 - \gamma)^2}\left(2\epsilon_\pi + \epsilon_\psi + \epsilon_\omega + \gamma\epsilon_\theta\right).$$

$\square$

**Theorem 3** (Branched rollout return bound for decentralized model) **.** *The gap between return in $n + 1$th policy rollout $\eta(\pi, \psi^{n+1})$ and return in $k$-step branched rollout with $h$-step experience with $n$th learned model $\eta^{branch}((\pi_D^n, \pi), (\psi^n, \psi_\omega^n))$ is bounded as:*

$$\left| \eta\left(\pi, \psi^{n+1}\right) - \eta^{branch}\left((\pi_D^n, \pi), (\psi^n, \psi_\omega^n)\right) \right| \leq C\left(\epsilon_\theta, \epsilon_\pi, \epsilon_\omega, \epsilon_\psi\right)$$

*Proof.*

$$\delta \triangleq \left| \eta\left(\pi, \psi^{n+1}\right) - \eta^{branch}\left((\pi_D^n, \pi), (\psi^n, \psi_\omega^n)\right) \right|$$

$$\leq \underbrace{\left| \eta\left(\pi, \psi^{n+1}\right) - \eta^{branch}\left((\pi_D^n, \pi_D^n), (\psi^n, \psi^n)\right) \right|}_{L_1}$$

$$+ \underbrace{\left| \eta^{branch}\left((\pi_D^n, \pi_D^n), (\psi^n, \psi^n)\right) - \eta^{branch}\left((\pi_D^n, \pi), (\psi^n, \psi_\omega^n)\right) \right|}_{L_2}$$

Apply Lemma 8 to $L_1$ and $L_2$:

$$L_1 \leq \frac{2r_{\max}}{1 - \gamma}\left[\left(h + \frac{\gamma^{h+k+1}}{1 - \gamma}\right)\left(\epsilon_\pi + \epsilon_\psi\right) + \left(\epsilon_\pi + \epsilon_\psi\right) + \gamma^{h+1}\left(k\left(\epsilon_\pi + \epsilon_\psi + \epsilon_\theta\right) + \epsilon_\pi + \epsilon_\psi\right)\right]$$

$$= \frac{2r_{\max}}{1 - \gamma}\left[\left(h + \frac{\gamma^{h+k+1}}{1 - \gamma} + 1 + (k+1)\gamma^{h+1}\right)\left(\epsilon_\pi + \epsilon_\psi\right) + k\gamma^{h+1}\epsilon_\theta\right],$$

$$L_2 \leq \frac{2r_{\max}}{1 - \gamma}\left[\left(h + \frac{\gamma^{h+k+1}}{1 - \gamma}\right)(\epsilon_\omega) + (\epsilon_\omega) + \gamma^{h+1}\left(k\left(\epsilon_\pi + \epsilon_\omega\right) + \epsilon_\pi + \epsilon_\omega\right)\right]$$

$$= \frac{2r_{\max}}{1 - \gamma}\left[\left(h + \frac{\gamma^{h+k+1}}{1 - \gamma} + 1\right)\epsilon_\omega + (k+1)\gamma^{h+1}\left(\epsilon_\pi + \epsilon_\omega\right)\right].$$

Thus,

$$\delta \leq \frac{2r_{\max}}{1 - \gamma}\left[\left(\frac{\gamma^{h+k+1}}{1 - \gamma} + h + 1 + (k+1)\gamma^{h+1}\right)\left(\epsilon_\pi + \epsilon_\psi + \epsilon_\omega\right) + k\gamma^{h+1}\left(\epsilon_\theta + k\epsilon_\pi\right)\right]$$

$\square$

**Theorem 4** (Rollout return bound for decentralized model with prediction error) **.** *The gap between return in $n + 1$th policy rollout $\eta(\pi, \psi^{n+1})$ and return in model rollout with $n$th learned model $\eta^{model}(\pi, \psi^n_\omega)$ is bounded as:*

$$\left| \eta\left(\pi, \psi^{n+1}\right) - \eta^{model}\left(\pi, \psi^n_\omega\right) \right| \leq \underbrace{\frac{2r_{\max}}{(1-\gamma)^2} \left(2\epsilon_\pi + \hat{\epsilon}_\omega + \gamma\epsilon_\theta\right).}_{C(\epsilon_\theta, \epsilon_\pi, \hat{\epsilon}_\omega)}$$

*Proof.*

$$\left| \eta\left(\pi, \psi^{n+1}\right) - \eta^{model}\left(\pi, \psi^n_\omega\right) \right| \leqslant \underbrace{\left| \eta\left(\pi, \psi^{n+1}\right) - \eta\left(\pi^n_D, \psi^n_\omega\right) \right|}_{L_1} + \underbrace{\left| \eta\left(\pi^n_D, \psi^n_\omega\right) - \eta^{model}\left(\pi, \psi^n_\omega\right) \right|}_{L_2}$$

Apply Lemma 7 to $L_1$ and $L_2$:

$$L_1 \leq \frac{2r_{\max}}{1-\gamma} \left( \frac{\gamma\left(\epsilon_\pi + \hat{\epsilon}_\omega\right)}{1-\gamma} + \epsilon_\pi + \hat{\epsilon}_\omega \right)$$

$$L_2 \leq \frac{2r_{\max}}{1-\gamma} \left( \frac{\gamma\left(\epsilon_\pi + \epsilon_\theta\right)}{1-\gamma} + \epsilon_\pi \right)$$

Thus,

$$\left| \eta\left(\pi, \psi^{n+1}\right) - \eta^{model}\left(\pi, \psi^n_\omega\right) \right| \leq \frac{2r_{\max}}{(1-\gamma)^2} \left(2\epsilon_\pi + \hat{\epsilon}_\omega + \gamma\epsilon_\theta\right).$$

$\square$

**Theorem 5** (Branched rollout return bound for decentralized model with prediction error) **.** *The gap between return in $n + 1$th policy rollout $\eta(\pi, \psi^{n+1})$ and return in $k$-step branched rollout with $h$-step experience with $n$th learned model $\eta^{branch}((\pi^n_D, \pi), (\psi^n, \psi^n_\omega))$ is bounded as:*

$$\left| \eta\left(\pi, \psi^{n+1}\right) - \eta^{branch}\left((\pi^n_D, \pi), (\psi^n, \psi^n_\omega)\right) \right| \leq C\left(\epsilon_\theta, \epsilon_\pi, \epsilon_\omega, \hat{\epsilon}_\omega\right).$$

*Proof.*

$$\delta \triangleq \left| \eta\left(\pi, \psi^{n+1}\right) - \eta^{branch}\left((\pi^n_D, \pi), (\psi^n, \psi^n_\omega)\right) \right|$$

$$\leq \underbrace{\left| \eta\left(\pi, \psi^{n+1}\right) - \eta^{branch}\left((\pi^n_D, \pi^n_D), (\psi^n_\omega, \psi^n_\omega)\right) \right|}_{L_1}$$

$$+ \underbrace{\left| \eta^{branch}\left((\pi^n_D, \pi^n_D), (\psi^n_\omega, \psi^n_\omega)\right) - \eta^{branch}\left((\pi^n_D, \pi), (\psi^n, \psi^n_\omega)\right) \right|}_{L_2}$$

Apply Lemma 8 to $L_1$ and $L_2$:

$$L_1 \leq \frac{2r_{\max}}{1-\gamma} \left[ \left(h + \frac{\gamma^{h+k+1}}{1-\gamma}\right)\left(\epsilon_\pi + \hat{\epsilon}_\omega\right) + \left(\epsilon_\pi + \hat{\epsilon}_\omega\right) + \gamma^{h+1}\left(k\left(\epsilon_\pi + \hat{\epsilon}_\omega + \epsilon_\theta\right) + \epsilon_\pi + \hat{\epsilon}_\omega\right) \right]$$

$$= \frac{2r_{\max}}{1-\gamma} \left[ \left(h + \frac{\gamma^{h+k+1}}{1-\gamma} + 1 + (k+1)\gamma^{h+1}\right)\left(\epsilon_\pi + \hat{\epsilon}_\omega\right) + k\gamma^{h+1}\epsilon_\theta \right],$$

$$L_2 \leq \frac{2r_{\max}}{1-\gamma} \left[ \left(h + \frac{\gamma^{h+k+1}}{1-\gamma}\right)\left(\epsilon_\omega\right) + \left(\epsilon_\omega\right) + \gamma^{h+1}\left(k\left(\epsilon_\pi\right) + \epsilon_\pi\right) \right]$$

$$= \frac{2r_{\max}}{1-\gamma} \left[ \left(h + \frac{\gamma^{h+k+1}}{1-\gamma} + 1\right)\epsilon_\omega + (k+1)\gamma^{h+1}\epsilon_\pi \right].$$

Thus,

$$\delta \leq \frac{2r_{\max}}{1-\gamma} \left[ \left(\frac{\gamma^{h+k+1}}{1-\gamma} + h + 1\right)\left(\epsilon_\pi + \hat{\epsilon}_\omega + \epsilon_\omega\right) + (k+1)\gamma^{h+1}\left(2\epsilon_\pi + \hat{\epsilon}_\omega\right) + k\gamma^{h+1}\epsilon_\theta \right]$$

$\square$

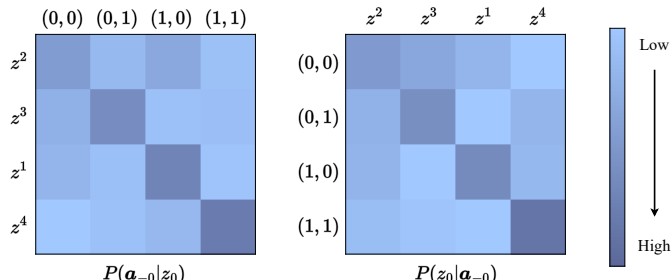

Figure 5: Conditional probability matrices: $P(\boldsymbol{a}_{-0}|z_0)$ (left) and $P(z_0|\boldsymbol{a}_{-0})$ (right). Deeper color means higher probability. The order of $z_0$ has been adjusted for clarification.

# B   VERIFICATION ON LEARNED LATENT VARIABLE

To examine how related the learned latent variable and the inaccessible information are, we designed a simple tabular case, where policies, transition matrix, and reward matrix are preset. There are 3 states and 3 agents with 2 actions for each and the space of latent variable is set to be 4. For agent 0, we collect experiences $\langle s, a_0, s', r \rangle$ and train a latent variable model end-to-end. For visualization, we design $\psi_\omega$ as an explicit network to fetch learned $z_0$ and preset the forward pass for the state in $P_\theta$ to avoid the correspondence being conditioned on the state. Then, we sample $z_0$ from learned latent variable function network $\psi_\omega$ for each experience in the buffer, and then calculate the conditional probabilities, $P(z_0|\boldsymbol{a}_{-0})$ and $P(\boldsymbol{a}_{-0}|z_0)$. As shown in Figure 5, there is a one-to-one correspondence between the latent variable and other agents' joint action. This demonstrates the latent variable model can implicitly capture inaccessible information relevant to transition and reward via end-to-end learning.

# C   EXPERIMENT DETAILS

## C.1   ENVIRONMENT SETTING

In this section, we introduce the environment settings we used in the experiments.

**Stochastic Game.** In our stochastic game, there are 30 observations, 3 agents and 5 actions for each agent, and episode length is limited to 40 steps. We generate a transition matrix $T$ and a reward matrix $R$ in advance as transition function and reward function. Concretely, $T$ is a matrix in shape of $[30, 5, 5, 5, 30]$ and $R$ is a matrix in shape of $[30, 5, 5, 5, 1]$. At each timestep $t$, given observation $o_t$ and agent joint actions $\left(a_t^0, a_t^1, a_t^2\right)$, the transition is:

$$o_{t+1} \sim T[o_t, a_t^0, a_t^1, a_t^2], \ r_t = R[o_t, a_t^0, a_t^1, a_t^2].$$

**MPE.** In our MPE tasks, agents observe their own positions, velocity, and others' relative positions. And actions of agents control their accelerations in every direction which is continuous in our experiments. In MPE tasks, the episode length is limited to 40 steps.

**4-Agent Cooperative Navigation.** In 4-agent Cooperative Navigation, as shown in Figure 6 (left), 4 agents learn to cooperate to reach 4 landmarks respectively. Concretely, we denote the radius of agent $i$ as $d^i$, position of agent $i$ as $(x_a^i, y_a^i)$ and position of landmark $i$ as $(x_l^i, y_l^i)$, and the reward is:

$$r = -\sum_{i=0}^{3} \left( \min_j \sqrt{(x_l^i - x_a^j)^2 + (y_l^i - y_a^j)^2} \right) - p_c,$$

where $p_c$ is a collision penalty:

$$p_c = \sum_{0 \leq i,j \leq 3} \left[ \mathbb{I}_{\{x|x<d_i+d_j\}} \left( \sqrt{(x_a^i - x_a^j)^2 + (y_a^i - y_a^j)^2} \right) \right].$$

Thus, the reward upper bound at each step is $-4$.

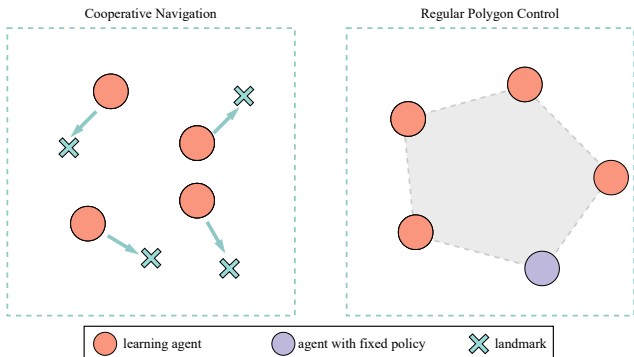

Figure 6: Illustration of MPE tasks: Cooperative Navigation (left) and Regular Polygon Control (right).

**5-Agent Regular Polygon Control.** In 5-agent Regular Polygon Control, as shown in Figure 6 (right), 4 agents learn to cooperate with another agent, which is controlled by a fixed policy, aiming to form a regular pentagon. The fixed policy is that the acceleration of the agent is always in the direction of the relative position between the center of the other 4 agents and itself. And the reward is given according to the area $S$ of current pentagon scaled by its perimeter $C$, which formally is:

$$S_{scaled} = \begin{cases} S \cdot (\frac{10}{C})^2, & \text{agents form a convex pentagon} \\ 0, & \text{otherwise,} \end{cases}$$

and represents the area of its similar pentagon with a perimeter of 10. So when the pentagon is a regular pentagon, $S_{scaled}$ comes to its maximum, $5 \cot \frac{\pi}{5}$. Additionally, two penalty items are given. Bound penalty, $p_b$, is used to restrict agents to stay in bounds :

$$bound(x) = \begin{cases} 0, & |x| < 0.9 \\ 10 * (|x| - 0.9), & |x| < 1.0 \\ e^{2|x|-2}, & \text{otherwise} \end{cases},$$

$$p_b = \sum_{i=0}^{3} \left[ bound(x^i) + bound(y^i) \right],$$

where $(x^i, y^i)$ is the position of agent $i$. Collision penalty, $p_c$, is as same as that in 4-agent Cooperative Navigation. Finally, we design the reward as:

$$r = \min\left\{ \max\left\{ S_{scaled}, \frac{1}{5 \cot \frac{\pi}{5} - S_{scaled}} \right\}, 1000 \right\} - 4p_c - p_b,$$

where $\max$ operator helps to distinguish when pentagon is relatively large and $\min$ operator handles the situation being divided by zero. This task is more difficult than Cooperative Navigation.

**Multi-Agent MuJoCo.** In our multi-agent MuJoCo experiments, the state in MuJoCo environment, which describes the position, velocity, angular velocity of each joint, etc, is used as the observation distributed to each agent. Specifically, in Ant task, we only use dimensions from 0 to 26 of the state. We limit the episode length of Halfcheetah to 250 steps, and 500 steps for Ant and Hopper. We provide the joint allocation of each task in Table 1.

### C.2 Implementation & Hyperparameters

In this section, we provide details for implementation and hyperparameters.

For the experiment environment, we adopt MPE (MIT license) and MuJoCo Gym (MIT license). For PPO, we follow the version in OpenAI's Spinning Up (MIT license).

Table 1: Joint allocation in multi-agent MuJoCo tasks. The relation column indicates how agents control the joints of robotics.

| Task | MuJoCo action | Multi-agent MuJoCo actions | Relation |
|---|---|---|---|
| Hopper 3×1 | $(a_0, a_1, a_2)$ | $[(a_0^0), (a_0^1), (a_0^2)]$ | $a_i = a_0^i$ |
| Ant 4×2 | $(a_0, \cdots, a_7)$ | $[(a_0^0, a_1^0), \cdots, (a_0^3, a_1^3)]$ | $a_{2i+k} = a_k^i$ |
| HalfCheetah 3×2 | $(a_0, \cdots, a_5)$ | $[(a_0^0, a_1^0), \cdots, (a_0^2, a_1^2)]$ | $a_{2i+k} = a_k^i$ |
| HalfCheetah 6×1 | $(a_0, \cdots, a_5)$ | $[(a_0^0), \cdots, (a_0^5)]$ | $a_i = a_0^i$ |
| HalfCheetah 5:[1,1,1,1,2] | $(a_0, \cdots, a_5)$ | $[(a_0^0), \cdots, (a_0^3), (a_0^4, a_1^4)]$ | $a_{i+k} = a_k^i$ |
| HalfCheetah 5×2 | $(a_0, \cdots, a_5)$ | $[(a_0^0, a_1^0), \cdots, (a_0^4, a_1^4)]$ | $a_i = a_0^i, a_5 = \frac{\sum_i a_1^i}{5}$ |

Table 2: Structure of the neural networks we used in experiments.

| Network | Stochastic Game | | MPE | | Multi-agent MuJoCo | |
|---|---|---|---|---|---|---|
| | hidden | activation | hidden | activation | hidden | activation |
| PPO Actor | (128,128) | tanh | (128,128) | tanh | (128,128) | tanh |
| PPO Critic | (128,128) | tanh | (128,128) | tanh | (128,128) | tanh |
| $\psi_\omega$ | (64,64) | ReLU | (64,64) | ReLU | (32,32) | ReLU |
| $f_\zeta$ | (128) | ReLU | (64,64) | ReLU | (256) | ReLU |
| $P_\theta$ | (128,64) | ReLU | (64) | ReLU | (128,64) | ReLU |
| $R_\phi$ | (128,64) | ReLU | (64,64) | ReLU | (128,64) | ReLU |

Table 3: Hyperparameters

| | Stochastic Game | MPE | Multi-agent MuJoCo |
|---|---|---|---|
| latent variable dimension | 3 | 4 | 6 |
| $\lambda$ | | 0.94 | |
| $\gamma$ | | 0.98 | |
| $\epsilon$ | | 0.2 | |
| $\epsilon_{value}$ | | 10 | |
| $c_{entropy}$ | | 3e-4 | |
| $c_{value}$ | | 0.5 | |
| max gradient norm | | 0.6 | |
| PPO batch size | | 32 | |
| actor learning rate | | 3e-4 | |
| critic learning rate | | 1e-3 | |
| $k$ | 4 | | 2 |
| $h$ | 8 | | 6 |
| $l$ | | 8 | |
| $c_{o\prime}$ | 10 | $1,100^4$ | 5 |
| latent variable model batch size | | 64 | |
| prediction batch size | 32 | | 128 |
| $\psi_\omega$ learning rate | | 1e-5 | |
| $f_\zeta$ learning rate | 1e-4 | 3e-5 | 5e-5 |
| $P_\theta$ learning rate | 1e-4 | | 3e-5 |
| $R_\phi$ learning rate | 1e-4 | | 3e-5 |

All neural networks used in our implementation are in the form of Multi-Layer Perception (MLP). Particularly, the transition function and reward function are respectively learned using an ensemble formed by 3 individual versions of the last layer. The likelihood of next observation is multiplied by a coefficient $c_{o\prime}$ to balance the scale of the elements in (3) and (4). The hidden size and activation function used in the networks are provided in Table 2. And the parameters used in training are provided in Table 3.

---

[4] $c_{o\prime} = 1$ in 4-Agent Cooperative Navigation task and $c_{o\prime} = 100$ in 5-Agent Regular Polygon Control

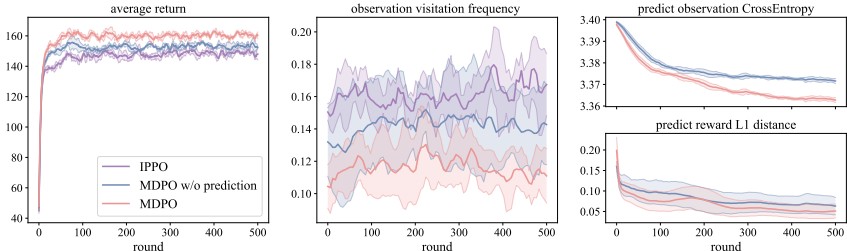

Figure 7: Learning curves of MDPO compared with MDPO w/o prediction and IPPO on the single-agent non-stationary stochastic game: average return (left), observation visitation frequency divergence (mid), and model prediction errors (right). Each round is 1600 environment steps.

In the implementation of latent variable function, both deterministic and stochastic latent variable satisfy our analysis, and we choose between them according to environment properties and experimental performance. In MPE and stochastic game except for Appendix B, we use deterministic latent variable with L2 regularization. In stochastic game in Appendix B, we use Category distribution. And in multi-agent MuJoCo environment, we use Gaussian distribution. As for the implementation of transition function and reward function, we use Category distribution for transition function in stochastic game and deterministic output for others.

The experiments are carried out on Intel i9-10900K CPU and NVIDIA GTX 3080Ti GPU. The training of stochastic game task costs 6 hours, while it takes 14 hours for each MPE task, and 25 hours for each multi-agent MuJoCo task.

## D ADDITIONAL RESULTS

Since MDPO helps to handle non-stationarity in decentralized MARL from the perspective of an individual agent, it will be natural and easy to also apply MDPO to single-agent RL in non-stationary environments. In this section, we investigate how MDPO performs in such a non-stationary single-agent environment.

We adopt the cooperative stochastic game into a single-agent non-stationary version. Concretely, we fix the policies of two agents and leave only one agent to update its policy. And we generate 5 noise matrices $(N_0, \cdots, N_4)$ randomly, which are in the same shape as the transition matrix $(T)$ and will influence the transition probability in a rotating manner. Formally, in $n$th policy rollout, the transition matrix is $T + N_{n \bmod 5}$, and we guarantee such a transition matrix is reasonable when generating noise matrices.

We compare the performance of MDPO, MDPO w/o prediction, and IPPO on the single-agent non-stationary stochastic game, and the learning curves are shown in Figure 7. As illustrated in Figure 7 (left), MDPO still performs the best in the single-agent non-stationary environment. As shown in Figure 7 (right), latent variable prediction helps to predict the non-stationary transition, and as no noise is applied to the reward matrix, there is merely a slight difference in reward prediction. Since the latent variable in this environment (noise matrices) is in a regular rotation, the prediction function is easier to learn than in decentralized MARL settings. However, unlike in decentralized MARL, non-stationarity in this setting will not fade away in pace with policy convergence, thus MDPO w/o prediction may keep oscillating and generate experiences with larger observation visitation frequency divergence than MDPO, which is shown in Figure 7 (mid).

Generally, MDPO also works in single-agent non-stationary environments, especially when there is a regular pattern of non-stationarity. More thorough studies are left as future work.

## E ADDITIONAL RELATED WORK

By utilizing an environment model, model-based RL has shown many advantages, such as sample efficiency (Wang et al., 2019) and exploration (Pathak et al., 2017). Many paradigms have been

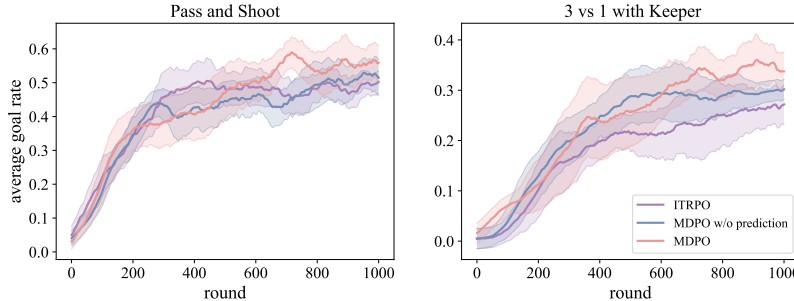

Figure 8: Learning curves of MDPO, MDPO w/o prediction, and ITRPO in Run and Pass (left) and 3 vs 1 with Keeper (right). Each round is 3200 environment steps.

proposed on how to exploit the environment model. Model-based planning methods, such as model predictive control, select actions through model rollouts. Dyna-style methods (Sutton, 1990; Feinberg et al., 2018; Janner et al., 2019) use both data collected in the real environment and data generated by the learned model to update the policy. Recent studies have extended model-based methods to multi-agent settings for sample efficiency in zero-sum game (Zhang et al., 2020), in stochastic game (Zhang et al., 2021) and in networked system (Du et al., 2022), centralized training (Willemsen et al., 2021), opponent modeling (Yu et al., 2021b), and communication (Kim et al., 2021). However, none of them strictly tackle fully decentralized setting of our paper.

Specially, the DMPO (decentralized model-based policy optimization) algorithm in prior work (Du et al., 2022) is designed for a networked system, where agents are able to communicate along the edges with their neighbors. The naming of *DMPO* and *MDPO* may lead to misunderstanding of similar settings, but the two algorithms are actually concerned with different settings. In fully decentralized setting of our paper, no information sharing is allowed between agents. And when the number of neighbors is set zero in DMPO, it will degenerate into the version of MDPO w/o prediction in our algorithm.

## F  ADDITIONAL EXPERIMENTS

In order to enrich our baseline and test MDPO on more complex environment, we supplement experiments on Google Research Football environment (GRF) with ITRPO as the baseline. ITRPO is the decentralized version of TRPO (Schulman et al., 2015), and we use TRPO to optimize the policy in MDPO and MDPO w/o prediction. Specifically, We choose 'simple115v2' as observation representation which encodes the state with 115 floats and 'scoring+checkpoint' as reward which encodes the domain knowledge that scoring is aided by advancing across the pitch. We examine MDPO, MDPO w/o prediction and ITRPO in two scenarios, Run and Pass and 3 vs 1 with Keeper, in both of which MDPO improves the average goal rate. The learning curves is shown in Figure 8.

In our implementation, we use Category distribution for one-hot dimensions in observation and Gaussion distribution for others to model the transition. We use MLP for reward function and Gaussion distribution for latent variable function. The structures are listed in Table 4 and the hyperparameters are listed in Table 5.

Table 4: Structure of the neural networks we used in GRF experiments.

| Network | Google Research Football | |
|---|---|---|
| | hidden | activation |
| TRPO Actor | (128,128) | tanh |
| TRPO Critic | (128,128) | tanh |
| $\psi_\omega$ | (128,128) | ReLU |
| $f_\zeta$ | (256) | ReLU |
| $P_\theta$ | (128,128) | ReLU |
| $R_\phi$ | (128,128) | ReLU |

Table 5: Hyperparameters in GRF experiments

| TRPO hyperparameters | | MDPO hyperparameters | |
|---|---|---|---|
| $\lambda$ | 1.0 | latent variable dimension | 6 |
| $\gamma$ | 0.99 | $k$ | 4 |
| KL limitation | 0.06 | $h$ | 16 |
| damping coefficient | 0.2 | $l$ | 8 |
| conjugate gradient iteration | 5 | $c_{o\prime}$ | 1 |
| backtrack iteration | 5 | latent variable model batch size | 128 |
| backtrack coefficient | 0.8 | prediction batch size | 256 |
| max gradient norm | 10 | $\psi_\omega$ learning rate | 1e-5 |
| TRPO batch size | 32 | $f_\zeta$ learning rate | 5e-5 |
| $\epsilon_{value}$ | 10 | $P_\theta$ learning rate | 3e-5 |
| critic learning rate | 5e-5 | $R_\phi$ learning rate | 1e-5 |

