# OpenReview forum: "Model-Based Decentralized Policy Optimization "
_ICLR.cc/2023/Conference — Submitted to ICLR 2023_

### Official Review · Reviewer_p6F5 · 2022-10-24

**Confidence:** 2
**Correctness:** 3
**Technical Novelty And Significance:** 3
**Empirical Novelty And Significance:** 3
**Recommendation:** 6

**Clarity, Quality, Novelty And Reproducibility:**


<Methodology>

1. The description of the structure of the encoder model that estimates the latent variable is poor. In particular, it would be nice if you authors explain what kind of distribution is used to model the latent variable and what architecture was used.

2. I am wondering if the authors have tried to use a variational learning approach to learn the encoder model capturing the latent factor. The variational approach can possibly prevent the overfitting of the latent variable function to certain datasets. In addition, the variational recurrent model can be possibly used to model both the latent factors and their temporal variation. This approach can be beneficial in that one can impose proper inductive biases regularizing the temporal behavior of latent factors, i.e., temporal consistency.

3. In Equation 3, it is strange to learn a transition model based on a latent variable and a reward function as one objective function. There must be a difference in scale between the elements constituting the objective function, but I wonder if the learning is going well.

4. How to compare the proposed approach with the other decentralized MARL approach modeling opponent behaviors and using them to optimize individual policy?

5. I am curious about the mechanism of cooperation. Although agents share the same reward, there is the chance that individually selected action does not necessarily induce cooperation.


<Experiments>

It may be required to include an additional baseline. For example, MARL with an opponent modeling approach.


**Strength And Weaknesses:**

1. This study explains in detail why decentralized MARL learning is difficult and explains in detail and theoretically the tasks that must be preceded in order to overcome the difficulties.

2. The performance bound of decentralized MARL learning was analyzed very strictly by correlating the performance of the prediction model.

3. The mathematical analysis is rigorous and detailed, but the description of the structure of the model used and the learning method is relatively week. In particular, reasons and justifications for using a specific model structure and choosing a learning method are not sufficiently provided.


**Summary Of The Paper:**

The study proposes a model-based decentralized policy optimization for MARL to overcome the non-stationary issues arising from the concurrent policy learning of multiple agents. Specifically, the current study proposes modeling individual agents’ state transition model and reward function while taking into account other agents’ policies into a latent variable. The modeled individual dynamic and reward functions along with the latent factor encoder are used to train an individual decision-making policy. Furthermore, to take into account the nonstationary aspect of other agents’ policies, the current study uses a latent variable prediction function.

**Summary Of The Review:**

The mathematical analysis is rigorous and detailed, but the description of the structure of the model used and the learning method is relatively week.

---

> ### Author Response · Authors · 2022-11-17
> **Response to Reviewer p6F5**
>
> We thank the reviewer for the insightful reviews on our work!
>
> > The description of the structure of the encoder model that estimates the latent variable is poor. In particular, it would be nice if you authors explain what kind of distribution is used to model the latent variable and what architecture was used.
>
> Thanks for pointing this out. Both deterministic and stochastic latent variable satisfies our analysis, and we choose from them according to environment properties and experimental performance. In MPE and stochastic game except for Appendix B, we use deterministic latent variable with L2 regularization. In stochastic game in Appendix B, we use Category distribution. And in Multi-Agent MuJoCo environment, we use Gaussian distribution.
>
> > I am wondering if the authors have tried to use a variational learning approach to learn the encoder model capturing the latent factor. The variational approach can possibly prevent the overfitting of the latent variable function to certain datasets. In addition, the variational recurrent model can be possibly used to model both the latent factors and their temporal variation. This approach can be beneficial in that one can impose proper inductive biases regularizing the temporal behavior of latent factors, i.e., temporal consistency.
>
> We have experimented on variational approach and found the current approach is empirically better than the variational version. So we choose the current approach.
>
> > In Equation 3, it is strange to learn a transition model based on a latent variable and a reward function as one objective function. There must be a difference in scale between the elements constituting the objective function, but I wonder if the learning is going well.
>
> Thanks for point this out. In fact, there is a coefficient to balance the two terms in implementation. We have made it clear in Appendix C.2.
>
> > How to compare the proposed approach with the other decentralized MARL approach modeling opponent behaviors and using them to optimize individual policy?
>
> > It may be required to include an additional baseline. For example, MARL with an opponent modeling approach.
>
> To the best of our knowledge, all of the existing opponent modeling methods require the information of other agents during training, and thus are not decentralized methods. We think it is unfair to compare MDPO with them. In order to enrich our experimental results, we have run experiments on Google Research Football (GRF) environment with ITRPO as a newly added baseline and we have supplemented the experimental results in Appendix F. MDPO improves the average goal rate on both of the scenarios we tested on.
>
> > I am curious about the mechanism of cooperation. Although agents share the same reward, there is the chance that individually selected action does not necessarily induce cooperation.
>
> Yes, this is possible during exploration. For example, for policy gradient methods, the action sampled from current policy can be a bad action. But, as agents share the same reward, the gradient update will decrease the probability of such actions, thus coordination is gradually promoted.

---

### Official Review · Reviewer_DZj6 · 2022-10-30

**Confidence:** 4
**Correctness:** 3
**Technical Novelty And Significance:** 2
**Empirical Novelty And Significance:** 2
**Recommendation:** 3

**Clarity, Quality, Novelty And Reproducibility:**

The reproducibility is difficult to evaluate as code is not provided. The optimization part is also not clear. See summary of the review below.


**Strength And Weaknesses:**

Strength：
– The paper is clearly motivated and well structured.
– Theories and proofs are provided to support the algorithm.

Weakness:
– The authors made good theory contributions. But how the theory would guide the design of the algorithm and how the experiments show this relation are not clear to me.
– the experiments can be enhanced by comparing to recent model-based RL methods.


**Summary Of The Paper:**

The main contribution of this paper is to propose MDPO, a model-based decentralized policy optimization (MDPO), which incorporates a latent variable function to help construct the transition and reward function from an individual perspective.  The theoretical monotonic improvement guarantee is given and experimental results show the superiority of the proposed method.


**Summary Of The Review:**

The problem considered in this paper is clearly motivated. I find the methodology lacks motivation and the relation between the theoretical results and algorithm design (sec 3.5) is not clearly addressed.

Some detailed comments are listed below.

– Can authors explain why the first two inequalities in sec 3 hold?

– In sec 3.1, the authors propose to maximize likelihood of the dynamic and reward prediction model. How is $P_\theta\left(o^{\prime} \mid o, a, z\right)$ implemented? Do you use MLP or gaussian distribution? Similarly, how do you implement $\left(R_\phi\left(o, a, o^{\prime}, z\right)\right.$?

– There is another work on decentralized model-based policy optimization (DMPO). The setting of DMPO can be equal to this work by setting the number of neighbors as zero.
Can the authors include this method as a baseline?

Yali Du, Chengdong Ma, Yuchen Liu, Runji Lin, Hao Dong, Jun Wang, and Yaodong Yang. Fully decentralized model-based policy optimization for networked systems. arXiv preprint arXiv:2207.06559, 2022.

– Line 9 of algorithm 1, how is $\psi_{w_{j+1}}$ updated?

– The authors mentioned that MDPO is a decentralization method, but in the process of learning hidden variables, it is actually necessary to obtain the information of other agents. How could the algorithm be decentralized in this sense?

---

> ### Author Response · Authors · 2022-11-17
> **Response to Reviewer DZj6**
>
> We thank the reviewer for the detailed review on our work! The main concerns about the connection between theory and algorithm design and a recent model-based method are addressed as follows, as well as other issues.
>
>
> > The authors made good theory contributions. But how the theory would guide the design of the algorithm and how the experiments show this relation are not clear to me.
>
> Our work aims to ameliorate the poor monotonic improvement guarantee of policy optimization methods in decentralized MARL setting, which results from the huge gap between $\rho^{\pi_{new}}$ and $\rho^{\pi_{old}}$. We bridge the gap between $\rho^{\pi_{new}}$ and $\rho^{\pi_{old}}$ by using an environmental model with latent variable function, **which has been theoretically analyzed in Section 3.2 and is exact MDPO w/o prediction algorithm**. Moreover, the return bound between policy rollout and model rollout should be controllable for truly monotonic improvement, and we found constraining the prediction error of latent variable helps to control the bound, **which has been theoretically analyzed in Section 3.3 and 3.4**. **Thus, we designed latent variable prediction mechanism where we derived the complete MDPO algorithm.**
>
> Furthermore, we have experimentally verified these two points in our experiments on Stochastic Game (Section 4.1). In detail, we directly computed the observation visitation frequency divergence to verify the benefit of using latent variable model. And we compared the observation and reward prediction error with or without prediction mechanism to verify our prediction mechanism does help to constrain the prediction error. We hope these explanations should have made them clear to the reviewer.
>
> These two points are also admitted by other reviewers.
> * reviewer WQD2:
>
>   > MDPO learns a latent variable to help distinguish different transitions resulting from varying unobservable information of the full state and other agents' policies. This novel approach makes the observation transition  and reward  models more stable during training.
>
> * reviewer p6F5:
>
>   > Furthermore, to take into account the nonstationary aspect of other agents’ policies, the current study uses a latent variable prediction function.
>
>   > This study explains in detail why decentralized MARL learning is difficult and explains in detail and theoretically the tasks that must be preceded in order to overcome the difficulties.
>
>   > The performance bound of decentralized MARL learning was analyzed very strictly by correlating the performance of the prediction model.
>
>
> > Can authors explain why the first two inequalities in sec 3 hold?
>
> In fact, the two inequalities in Section 3 are used to illustrate our motivation before we carry out theoretical analysis. More concretely, inequality ① is what we discussed in Section 3.2 while inequality ② corresponds to Section 3.3 and 3.4. Hope this wouldn't cause any misunderstanding.
>
>
> > In sec 3.1, the authors propose to maximize likelihood of the dynamic and reward prediction model. How is $P_\theta(o\prime|o,a,z)$ implemented? Do you use MLP or gaussian distribution? Similarly, how do you implement $R_\phi(o,a,o\prime,z)$?
>
> Thanks for pointing this out. Both MLP and Gaussian distribution satisfy our analysis. In stochastic game, we use Category distribution for $P_\theta$ and MLP for $R_\phi$. In MPE and Multi-Agent MuJoCo environment, we use MLP for $P_\theta$ and $R_\phi$. We have made it more detailed in Appendix C.2.
>
>
> > There is another work on decentralized model-based policy optimization (DMPO). The setting of DMPO can be equal to this work by setting the number of neighbors as zero. Can the authors include this method as a baseline?
>
> This work was indeed included in Appendix E.
> When the number of neighbors is set as zero in DMPO, the algorithm will degenerate into the version of MDPO w/o prediction in our algorithm and we have compared MDPO with MDPO w/o prediction in the paper. We have elaborated this in Appendix E.
>
> In order to enrich our experimental results, we have run experiments on Google Research Football (GRF) environment with ITRPO as a newly added baseline and we have supplemented the experimental results in Appendix F. MDPO improves the average goal rate on both of the scenarios we tested on.

---

> > ### Author Response · Authors · 2022-11-17
> > **Response to Reviewer DZj6**
> >
> > > Line 9 of algorithm 1, how is $\psi_{\omega_{j+1}}$ updated?
> >
> > In line 9 of algorithm 1, we use '←' to illustrate the slide of the window of experience buffers and learned latent variable functions, as we maintain the $l$ **latest** learned latent variable functions and policy rollout experience buffers. Hope this wouldn't cause any misunderstanding.
> >
> > > The authors mentioned that MDPO is a decentralization method, but in the process of learning hidden variables, it is actually necessary to obtain the information of other agents. How could the algorithm be decentralized in this sense?
> >
> > As we stated in Section 3.1, we learn such a model by maximizing the likelihood of experiences of policy rollout D,$ \underset{\theta,\omega,\phi}{\max} \ \mathbb{E}_{(o,a,o^\prime,r)\sim\mathcal{D},z\sim\psi_\omega(\cdot|o)}\left[P_\theta\left(o^\prime|o,a,z\right)-(R_\phi\left(o,a,o^\prime,z\right) - r)^2\right] $
> >
> > **We train the latent variable function end-to-end without any information from other agents, which satisfies the setting of fully decentralized MARL.**

---

> ### Author Response · Authors · 2022-11-24
> **Follow-up**
>
> Dear reviewer:
>
> We would like to know whether our response have addressed your main concerns. If there are any additional questions about the paper, please let us know so that we can deal with it as early as possible. As you can see, your response is vital to us. We're looking forward to your feedback.
>
> Best Regards.

---

### Official Review · Reviewer_WQD2 · 2022-11-04

**Confidence:** 3
**Clarity, Quality, Novelty And Reproducibility:** Good
**Correctness:** 3
**Technical Novelty And Significance:** 3
**Empirical Novelty And Significance:** 2
**Recommendation:** 5

**Strength And Weaknesses:**

Strength:

1. MDPO learns a latent variable to help distinguish different transitions resulting from varying unobservable information of the full state and other agents' policies. This novel approach makes the observation transition $P_i(o_i'|o_i,a_i,z_i)$ and reward $R_i(o_i,a_i,z_i)$ models more stable during training.
2. This paper also provides theoretical performance bounds (Theorem 2, Theorem 3) for the proposed algorithm, which theoretically contributes to the monotonic policy improvement.

Weaknesses:

1. It needs to be clarified how MDPO solves the main problem of decentralized MARL. It seems that the main reason that makes $\|\rho^{\pi^\text{new}}-\rho^{\pi^\text{old}}\|>\|\rho^{\pi^\text{model}}-\rho^{\pi^\text{old}}\|$ hold is that the $\psi^n_w=(1-\alpha)\psi^{n-1}_w+\alpha\psi^n$ in Theorem 1, where $\psi^n$ is the true latent variable function. MDBP does not update $\psi$ directly to the true value, but uses a $\alpha$-tradeoff between the true value and the old value. Then does it mean that MDBP solves this problem by slightly updating each agent's policy like TRPO? As this paper says, such a slight update will lead to much slower convergence, especially in fully decentralized MARL. This is a key point about whether MDPO truly solves the nonstationary problem caused by the simultaneous update of policies (which also makes the assumption in TRPO not hold).
2. The experimental part of this paper needs to be more convincing. The baseline algorithm is only IPPO, and complex multi-agent environments such as The StarCraft Multi-Agent Challenge or Google Research Football are missing.

**Summary Of The Paper:**

This paper introduces a new approach (MDPO) to model-based decentralized reinforcement learning by leveraging a latent variable function $\psi(z|o)$ to help learn the observation transition $P_i(o_i'|o_i,a_i,z_i)$ and reward $R_i(o_i,a_i,z_i)$ models.

The main problem this paper tries to address is that the assumptions of TRPO may not hold in a decentralized MARL because multiple agents update their policies simultaneously, making a significant difference between the new and old joint policies. MDPO uses the environment model to bridge the gap between $\rho^{\pi^\text{new}}$ and $\rho^{\pi^\text{old}}$.

This paper also provides the proposed algorithm's theoretical performance bounds (Theorem 2, Theorem 3), which theoretically contribute to the monotonic policy improvement.

**Summary Of The Review:**

This paper introduces a latent variable to help learn the observation transition $P_i(o_i'|o_i,a_i,z_i)$ and reward $R_i(o_i,a_i,z_i)$ models. This paper proposes a new decentralized MARL algorithm, MDRL, to solve the nonstationary problem in a model-based way. However, I am negative about this paper based on the weak points above.

---

> ### Author Response · Authors · 2022-11-17
> **Response to Reviewer WQD2**
>
> We thank the reviewer for the detailed review on our paper. In response to the valuable comments, we would like to add the following details.
>
> > It needs to be clarified how MDPO solves the main problem of decentralized MARL. It seems that the main reason that makes $\|\rho^{\pi^{new}}-\rho^{\pi^{old}}\| > \|\rho^{\pi^{model}}-\rho^{\pi^{old}}\|$ hold is that the  $\psi_\omega^n=(1−\alpha) \psi_\omega^n+\alpha \psi^n$ in Theorem 1, where $\psi^n$ is the true latent variable function. MDBP does not update $\psi$ directly to the true value, but uses a $\alpha$-tradeoff between the true value and the old value. Then does it mean that MDBP solves this problem by slightly updating each agent's policy like TRPO? As this paper says, such a slight update will lead to much slower convergence, especially in fully decentralized MARL. This is a key point about whether MDPO truly solves the nonstationary problem caused by the simultaneous update of policies (which also makes the assumption in TRPO not hold).
>
> Basically, the update of $\psi$ and the update of other agents' policies are different. $\psi$ is the true latent variable function, and the update of $\psi$ depends on the update of other agents' policies. But when we consider the learned latent variable function $\psi_\omega$, the update of $\psi_\omega$ is controlled by the learning strategy. On one hand, the soft-update form of  $\psi_\omega$ will not hinder the policy update magnitude of other agents, which also satisfies the setting of fully decentralized MARL. On the other hand, the soft-update form also ensures experiences from several recent policy rollouts are used to update $\psi_\omega$ for sample efficiency, as mentioned in footnote 2 of our paper. Admittedly, using such a learned latent variable function $\psi_\omega$ is not entirely without cost. When there is a big update of true latent variable function $\psi$, the return gap between policy rollout and model rollout is large, resulting in invalid policy update from model rollout experiences. Thus, we use latent variable prediction mechanism to tradeoff between them.
>
> > The experimental part of this paper needs to be more convincing. The baseline algorithm is only IPPO, and complex multi-agent environments such as The StarCraft Multi-Agent Challenge or Google Research Football are missing.
>
> Thanks for pointing this out. To the best of our knowledge, our work is the first model-based method for fully decentralized MARL. And IPPO is a typical decentralized policy gradient algorithm.In order to enrich our experimental results, we have run experiments on Google Research Football (GRF) environment with ITRPO as a newly added baseline and we have supplemented the experimental results in Appendix F. MDPO improves the average goal rate in both Pass and Shot and 3 vs 1 with Keeper. As for StarCraft Multi-Agent Challenge, we explained in Section 4 that `We do not consider StarCraft multi-agent challenge (SMAC) (Samvelyan et al., 2019), because IPPO has been shown to perform very well in SMAC (de Witt et al., 2020; Papoudakis et al., 2021), close enough to centralized training with decentralized execution methods like QMIX (Rashid et al., 2018) and MAPPO (Yu et al., 2021a). Thus, the gain of MDPO may not be clearly evidenced there.`
>
> Thank the reviewer again for the meticulous and professional review work. We are looking forward to new comments.

---

> > ### Comment · Reviewer_WQD2 · 2022-11-18
> > **Comments to authors' response**
> >
> > Thanks for the authors' reply to my questions and for adding the experiments of MDPO and ITRPO in two scenarios of GRF.
> >
> > However, the author's response to my first question did not convince me. As the reply says,
> > > Basically, the update of $\psi$ and other agents' policies are different.
> >
> > It is right, but in model-based RL, the model's accuracy can significantly affect the update of the policy. Unfortunately, achieving good model calibration is difficult in a multi-agent context.
> >
> > From the presentation in the paper, $\psi$ attempts to represent the latent information (e.g., the policies of other agents) that cannot be observed by a particular agent in a partial observation multi-agent environment.
> > The authors use Eq. (4) to update the transition model, the model of rewards, and the $\psi$ function simultaneously.
> > Based on Eq. (4), $\phi$ could be viewed as an encoding layer for observation shared by the transition and reward models.
> >
> > Even assuming that $\phi$ is able to represent the unobservable part accurately, MDPO yields a policy $\pi^\text{model}$ with a smaller update magnitude than the original TRPO update $\pi^\text{new}$.
> > This contradicts the authors' aim of trying to improve the slow update drawback of TRPO.
> > More often than not, the inaccurate model makes the already uncertain external environment more unstable due to partial observation, which should be detrimental to the monotonic improvement of the policy.
> >
> > Despite the author's emphasis on
> > > To the best of our knowledge, our work is the first model-based method for fully decentralized MARL.
> >
> > I cannot find the benefit of introducing a model-based approach to the full decentralized paradigm.
> >
> > I agree with the authors' explanation on SMAC and also find the experimental results on GRF that the authors added in Appendix F. Unfortunately, MDPO does not seem to have a significant performance improvement. This may be due to the difficulty in learning an accurate state transfer model for complex environments.
> >
> > For the above reasons, I still feel negative about this manuscript.

---

> > > ### Author Response · Authors · 2022-11-19
> > > **Response to Reviewer WQD2**
> > >
> > > We thank the reviewer for the reply. But we think there may still be some misunderstandings and we want to make further explanation.
> > >
> > > > Based on Eq. (4), $\psi$ could be viewed as an encoding layer for observation shared by the transition and reward models.
> > >
> > > As the reviewer said in initial comments,
> > >
> > > > MDPO learns a latent variable to help **distinguish different transitions** resulting from varying unobservable information of the full state and other agents' policies.
> > >
> > > And we have experimentally verified that the learned latent variable is indeed related to the inaccessible information in Appendix B.
> > >
> > > > Even assuming that $\psi$ is able to represent the unobservable part accurately, MDPO yields a policy $\pi^{model}$ with a smaller update magnitude than the original TRPO update $\pi^{new}$.
> > >
> > > In fact, MDPO learns the model from **several** recent policy rollouts, which doesn't require smaller policy update magnitude waiting for model learning to catch up. And as we have stated in response, we use the latent variable prediction mechanism to **tradeoff** **between the stationarity of $\psi_\omega$ and the distance of $\psi$ and $\psi_\omega$**, which is verified in our experimental results in Section 4.1. As shown in Figure 2, in compared with MDPO w/o prediction, MDPO maintains relative stationary observation visitation frequency and obtains much lower prediction error with the prediction mechanism.
> > >
> > > > This contradicts the authors' aim of trying to improve the slow update drawback of TRPO.
> > >
> > > We didn't aim to improve the slow update of TRPO. It's just an intuitive solution to non-stationarity problem and it's obviously unacceptable. In fact, Our work aims to ameliorate the poor monotonic improvement guarantee of policy optimization methods in decentralized MARL setting, which results from the huge gap between $\rho^{\pi_{new}}$ and $\rho^{\pi_{old}}$. And we did not constraint TRPO/PPO at a slower update magnitude in our experiments.
> > >
> > > > Unfortunately, MDPO does not seem to have a significant performance improvement. This may be due to the difficulty in learning an accurate state transfer model for complex environments.
> > >
> > > Unfortunately, the GRF environment is not indeed a good environment for model learning due to its sparse reward setting. Even in such a tough environment for reward function learning, we do not think our improvement is trivial enough to ignore.
> > >
> > > |                    | MDPO           | MDPO w/o prediction | ITRPO      |
> > > | ------------------ | -------------- | ------------------- | ---------- |
> > > | Pass and Shoot     | **0.55±0.055** | 0.51±0.047          | 0.48±0.039 |
> > > | 3 vs 1 with Keeper | **0.34±0.041** | 0.29±0.021          | 0.26±0.038 |

---

> ### Author Response · Authors · 2022-11-24
> **Follow-up**
>
> Dear reviewer:
>
> We would like to know whether our response have addressed your main concerns. If there are any additional questions about the paper, please let us know so that we can deal with it as early as possible. As you can see, your response is vital to us. We're looking forward to your feedback.
>
> Best Regards.

---

### Decision · Program_Chairs · 2023-01-20

**Decision:**

Reject

**Justification For Why Not Higher Score:**

- Concerns about experimental evaluation
- Theoretical results are not strong enough to justify publication on their own

**Justification For Why Not Lower Score:**

N/A

**Metareview: Summary, Strengths And Weaknesses:**

This paper considers decentralized RL. A challenge in this setting is that, as an agent learns, other agents are updating their policies. This changes the distribution of the environment as observed by the agent in question, making it non-stationary. As a result, policy improvement methods may not necessarily provide monotone improvements over time.

The paper argues that building a model of reality is useful because it allows an agent to learn an optimal policy for this model without needing to deal with nonstationarity. In this setting, a model is introduced that uses a latent variable to represent the influence of other agents' policies on state-transitions and rewards.

A bound is shown (Theorem 1) suggesting that visitation frequencies under policy rollout for a specific model don't change too quickly as other agents update their policies. This is augmented by bounds (Theorem 2-6) on gaps between the policy rollout using the true latent variable and rollout using the learned latent variable. Although it is not formally stated or shown theoretically, intuition is given that this should result in monotone policy improvement.

Strengths
- Decentralized RL is an important and challenging problem
- It does seem plausible that using a model could allow agents to converge to good equilibria using fewer real-world trajectories
- The proposed algorithm is supported via theoretical bounds on the error in policy rollout between using the correct latent variable and an estimated one
- The approach is novel

Weaknesses
- There is a concern that inaccurate models can create a disadvantage that more than offsets the benefits of using a model to combat slow learning created by nonstationarity in other agents' behavior (reviewer WQD2)
- In the original experiments, only one baseline algorithm was used (IPPO) and more complex multi-agent environments were not included. When Google Research Football with a new baseline (ITRPO) was added during the rebuttal period, the proposed method offered a small but not statistically significant improvement over the new ITRPO baseline. (reviewer WQD2)
- The description of how the latent variable is estimated is not sufficiently detailed, making it difficult to judge its appropriateness, as articulated by Reviewer p6F5
- While somewhat helpful, the AC found that the theoretical results shown did not go as far as desired in demonstrating that the algorithm performs well. For example, claims about monotone policy improvement that are made supported by intuition are not formally stated. From the theoretical analysis, it isn't clear that the algorithm converges.